# Predicting multiple observations in complex systems through low-dimensional embeddings

Tao Wu [1], Xiangyun Gao [2,3] ✉, Feng An [4] ✉, Xiaotian Sun[2], Haizhong An[2,3], Zhen Su [5,6], Shraddha Gupta [5,7], Jianxi Gao [8,9] ✉ & Jürgen Kurths [5,7] ✉

Forecasting all components in complex systems is an open and challenging task, possibly due to high dimensionality and undesirable predictors. We bridge this gap by proposing a data-driven and model-free framework, namely, feature-and-reconstructed manifold mapping (FRMM), which is a combination of feature embedding and delay embedding. For a high-dimensional dynamical system, FRMM finds its topologically equivalent manifolds with low dimensions from feature embedding and delay embedding and then sets the low-dimensional feature manifold as a generalized predictor to achieve predictions of all components. The substantial potential of FRMM is shown for both representative models and real-world data involving Indian monsoon, electroencephalogram (EEG) signals, foreign exchange market, and traffic speed in Los Angeles Country. FRMM overcomes the curse of dimensionality and finds a generalized predictor, and thus has potential for applications in many other real-world systems.

Prediction of future states of a complex dynamical system is a challenging task across various disciplines[1–3]. System details are often unknown, and only their time series data are accessible. Therefore, a variety of data-driven techniques are designed for the prediction task[4,5], including traditional statistical models (e.g., autoregressive integrated moving average (ARIMA))[6], state space-based methods (e.g., sequential locally weighted global linear maps (S-maps)[7] and multiview embedding (MVE))[8], machine learning algorithms (e.g., support vector machine (SVM)[9], long short-term memory (LSTM)[10], and reservoir computing (RC)[11,12], and state-of-the-art combination frameworks (e.g., multitask learning-based Gaussian process regression machine (MT-GPRM)[13], randomly distribution embedding (RDE)[14] and autoreservoir neural network (ARNN)[15]. These advanced approaches have shown potential for several significant tasks, e.g.,

one-step and multistep ahead predictions of a target time series variable[16].

Despite considerable efforts in the study of prediction tasks in complex systems, it is still unsolved to design a generalized framework for the predictions of all components in a complex system. Since real-world systems often consist of many interconnected units, e.g., multiple spatiotemporal observations in climate systems[17] and thousands of functionally connected neurons in the brain[18], they therefore output a large number of time series variables, and the interactions between these variables intrinsically contribute to the dynamical evolution of a complex system. A practical way to predict complex systems (especially for high-dimensional systems), as an approximation, is to study the dynamics of partial units, e.g., representative observations[19]. However, identifying such representative variables remains a

[1]College of Management Science, Chengdu University of Technology, Chengdu 610059, China. [2]School of Economics and Management, China University of Geosciences, Beijing 100083, China. [3]Key Laboratory of Carrying Capacity Assessment for Resource and Environment, Ministry of Land and Resources, Beijing 100083, China. [4]School of Economics and Management, Beijing University of Chemical Technology, Beijing 100029, China. [5]Potsdam Institute for Climate Impact Research (PIK)–Member of the Leibniz Association, Potsdam 14473, Germany. [6]Department of Computer Science, Humboldt University at Berlin, Berlin 12489, Germany. [7]Department of Physics, Humboldt University at Berlin, Berlin 12489, Germany. [8]Department of Computer Science, Rensselaer Polytechnic Institute, Troy, NY 12180, USA. [9]Network Science and Technology Center, Rensselaer Polytechnic Institute, Troy, NY 12180, USA. ✉e-mail: gxy5669777@126.com; af15910602135@126.com; gaoj8@rpi.edu; kurths@pik-potsdam.de

challenging task. Moreover, one should be cautious to ignore those 'unimportant variables', in which small perturbations may be amplified and propagated to all components, resulting in heavy changes in system behaviors (known as cascading effects)[20,21]. Instead, the capacity to predict the future states of all components can help to better estimate the future behavior of a complex system. However, many existing approaches present typical limitations for this task. (a) The uncertainty of the predictor, which means that for a target variable, the predictors are often selected empirically[22], e.g., several target-related observations. If regarding all the remaining variables as predictors, some redundant information may negatively affect the performance (e.g., noise and irrelevant variables to the target variable[15]), especially for high-dimensional real-world systems[8]. (b) The uncertainty of the predictive model, which means that for different targets, some approaches may train different models (e.g., the predictive models for $y_1$ and $y_2$ are completely independent)[23,24]. There may need $N$ models for an $N$-dimensional system, leading to a high computational cost. (c) The challenge in forecasting multiple observations typically results in verifying methods over only a single or possibly few observations[13,14,23]. Therefore, designing a unified and reliable framework to forecast all components in complex systems is still an open and challenging issue.

In this work, we develop a data-driven and model-free framework by a combination of manifold learning and delay embedding, namely, feature-to-reconstructed manifold mapping (FRMM). The FRMM framework yields reliable predictions for all components via a generalized and practical predictor, i.e., the system's low-dimensional representation from manifold learning (feature embedding). The theoretical foundation of the FRMM is based on the ground truth that high-dimensional systems often contain redundant information and that their essential dynamics or structures can be characterized by low-dimensional representations[25–27], e.g., the meaningful structure of a 4096-dimensional image (64 pixels by 64 pixels) can be characterized in a three-dimensional manifold with two pose variables and an azimuthal lighting angle[28]. These low-dimensional representations can be sufficiently identified from two powerful techniques: feature embedding and delay embedding. (a) Feature embedding finds a low-dimensional representation by preserving the geometric features (e.g., nearest neighbor) of the original system as much as possible[25]. (b) Delay embedding reconstructs an isomorphic structure with the original system from a single time series[29]. Given that low-dimensional representations (in different coordinates) from two approaches show isomorphic structures with the original system. This enables prediction tasks by one-to-one mapping between two low-dimensional representations. Additionally, in a dynamical system, each time series variable can reconstruct a low-dimensional representation via delay embedding[30]. Therefore, the low-dimensional representation from feature embedding can be practically selected as a generalized predictor to potentially identify the future dynamics of all components in complex systems.

## Results

### Low-dimensional representation from delay embedding

According to Takens' embedding theory, it is possible to reconstruct a low-dimensional attractor by a single time series from a high-dimensional dynamical system[30]. Particularly, for an $N$-dimensional system $M$, one can reconstruct a topologically isomorphic manifold $M_{x_i}$ (namely, reconstructed manifold) from every time series $x_i(t)$ within the system ($i = 1, 2, \cdots, N$, $t = 1, 2, \cdots, L$, $L$ is the length of the series), and each state point on $M_{x_i}$ is represented as $\tilde{X}_i(t) = (x_i(t), x_i(t + \tau), \cdots, x_i(t + (E − 1)\tau))$, where $E$ is the embedding dimension and $\tau$ is the time lag. For example, the attractors of the 3-dimensional Lorenz system and Rössler system are reconstructed in 2-dimensional space from individual time series (Fig. 1a, b, d, e).

$M_{x_i}$ has an isomorphic topological structure with the original system $M$. It indicates that for every state point $X(t)$ on $M$, one can find a corresponding state point $\tilde{X}_i(t)$ on $M_{x_i}$ through a smooth mapping $\varphi_i$. According to Takens[30], $\varphi_i$ is a one-to-one mapping, we therefore identify a corresponding state point $X(t)$ on $M$ for every $\tilde{X}_i(t)$ on $M_{x_i}$ via the inverse mapping $\varphi^{-1}(\tilde{X}_i(t))$. These processes can be represented as (1).

$$\varphi_i : M \rightarrow M_{x_i}, \varphi_i(X(t)) = \tilde{X}_i(t), \varphi_i^{-1}(\tilde{X}_i(t)) = X(t), \quad (1)$$

where $X(t) = (x_1(t), x_2(t), \cdots, x_N(t))$ and $X(t) \in M, \tilde{X}_i(t) \in M_{x_i}$.

### Low-dimensional representation from feature embedding

Delay embedding can reconstruct low-dimensional representations of the original systems. Additionally, such low-dimensional

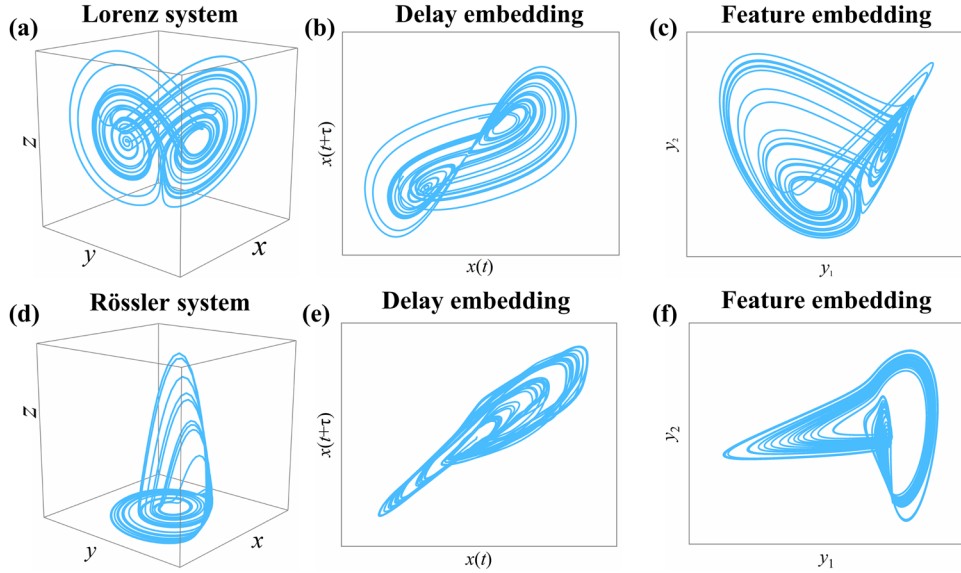

**Fig. 1 | Low-dimensional embeddings of complex systems.** The dynamical structure of the 3-dimensional Lorenz system (**a**) is represented in 2-dimensional space via delay embedding (**b**) (from time series $x$, where $E = 2, \tau = 10$) and feature embedding (**c**) (i.e., diffusion map algorithm). Analogously, one can find 2-dimensional representations (**e** and **f**) of the 3-dimensional Rössler system (**d**). Even with additive noise, one can also find their low-dimensional embeddings (cf. SI Appendix Fig. S1).

representations can be obtained from manifold learning algorithms. For example, based on the diffusion map algorithm[31,32], we find 2-dimensional representations that show equivalent structures with the 3-dimensional Lorenz and Rössler systems (Fig. 1c, f). These techniques embed a high-dimensional system in a low-dimensional space by retaining the essential geometric features (e.g., the neighboring points in high-dimensional space are also adjacent in the low-dimensional representations) (feature embedding) of the original system. Since the embedding is a one-to-one mapping[31], it can be written as (2)

$$\phi : M \rightarrow M_0, \phi(X(t)) = Y(t), \phi^{-1}(Y(t)) = X(t), \quad (2)$$

where $M_0$ represents an $E$-dimensional manifold (namely, feature manifold), and $Y(t) \in M_0, X(t) \in M$. Real-world systems often show diverse dynamical structures and geometry features, and five algorithms are selected alternatively to identify feature embedding, i.e., isometric feature mapping (ISOMAP)[28], locally linear embedding (LLE)[33], Laplacian[34], diffusion map[31], and local tangent space alignment (LTSA)[35]. More details are provided in Methods.

## Prediction via mapping between low-dimensional representations

Through delay embedding and feature embedding, a high-dimensional system is represented by its two low-dimensional manifolds: the reconstructed manifold ($M_x$) and the feature manifold ($M_0$). This indicates a one-to-one mapping between the feature manifold and the reconstructed manifold. Then, for every state point $Y(t)$ on $M_0$, we can find its corresponding state point $\tilde{X}_i(t)$ on $M_{x_i}$ by a smooth mapping (e.g., $\psi_i$).

$$\psi_i : M_0 \rightarrow M_{x_i}, \psi_i(Y(t)) = \tilde{X}_i(t), i = 1, 2, \cdots, N, \quad (3)$$

where $\psi_i(x) = \varphi_i \phi^{-1}(x)$ (see Eqs. (1) and (2)).

Note that $\tilde{X}_i(t) = (x_i(t), x_i(t+\tau), \cdots, x_i(t+(E-1)\tau))$, we deduce a spatiotemporal transformation from state points on $M_0$ to a temporal series (final component in $\tilde{X}_i(t)$):

$$\widehat{\psi_i}(y_1(t), y_2(t), \cdots, y_E(t)) = x_i(t+(E-1)\tau), i = 1, 2, \cdots, N, t = 1, 2, \cdots, L, \quad (4)$$

where $(y_1(t), y_2(t), \cdots, y_E(t)) \in M_0$ and $x_i(t+(E-1)\tau) \in \tilde{X}_i(t) \in M_{x_i}$. When $t = L$, it yields at most $(E-1)\tau$-forward dynamics of each variable $x_i(t)$ once $\widehat{\psi_i}$ is identified (more details of $\widehat{\psi_i}$ are provided in Methods). In this work, we employ the classical Gaussian process regression to train every $\widehat{\psi_i}$ (cf. SI Appendix Chapter 1.3)[36]. To guarantee robustness, we validate the performance by randomly dividing the observed series into a training set and a test set (i.e., cross-validation). Two widely used metrics are employed to measure the performance, i.e., the Pearson correlation between observed values and predicted values ($\rho$) and the normalized root mean square error (RMSE) (normalized by the standard deviation of input series)[15]. The main architecture of FRMM is given in Fig. 2.

## Performance of FRMM on model systems

To illustrate the mechanism of the FRMM framework, we start with the benchmark Lorenz system[15]. For the 3-dimensional ordinary Lorenz system (see Methods), the FRMM first identifies its 2-dimensional manifolds (i.e., feature manifold and reconstructed manifold) via feature embedding and delay embedding (Fig. 1a–c). Both feature manifolds and reconstructed manifolds show isomorphic structures with the original system, which indicates an isomorphism between the feature manifold and reconstructed manifold. Then, the feature manifold can be utilized as a generalized predictor for the predictions of three units. According to the generic cross-validation (see Methods), we validate the performance by randomly selecting 50% of the

data as training samples, and the others are test samples. Since the embedding dimension and time lag are $E = 2$ and $\tau = 10$, FRMM yields reliable 10-step ($T = (E-1)\tau$) ahead predictions for all units, where the average $\rho$ reaches 0.91 and the average error remains at a low level ($RMSE = 0.38$). Still, FRMM achieves accurate 10-step-ahead predictions for all units in the Rössler system (SI Appendix Fig. S2).

To further evaluate the FRMM framework in high-dimensional systems, we select the 90-dimensional coupled Lorenz system as a benchmark (see Methods)[14]. Traditional regression-based predictions encounter the "curse of dimensionality". Some neural network-based frameworks set all the observations as input, leading to relatively high computational costs. Due to the sensitivity of initial states as well as complex nonlinear dynamics between units, predicting all components is indeed a challenging task. Our FRMM framework embeds the 90-dimensional Lorenz system into a relatively lower space ($E = 11$) via feature embedding and delay embedding and sets the feature manifold as a generalized predictor to find the future states of all components (Fig. 3d–f). FRMM performs reliably in that all the $\rho$ values are higher than 0.6 and all the errors are lower than 0.8 (Fig. 3f). The average $\rho$ reaches 0.74, and the average RMSE is 0.6.

Real-world systems are often influenced by various external factors, and they may behave with time-dependent dynamics, e.g., the couplings among components are not constant but time-varying. For this case, we evaluate the FRMM by setting the coupling in the 90-dimensional Lorenz system to be increased by 0.2 after ten time intervals (see Methods)[15]. FRMM remains reliable with an average $\rho$ of 0.73 and RMSE of 0.63 (Fig. 3g–i).

## Performance of FRMM on real-world systems

To illustrate the FRMM in real-world systems, we use several benchmark samples across different disciplines, including climate systems, neuroscience, financial systems, and traffic systems (details of all datasets are provided in SI Appendix Chapter 1.2). a) For the climate system, we consider the Indian monsoon, which is a typical phenomenon that generates dramatic influences on India's agriculture and economy[37]. Skillful ahead prediction of this phenomenon is of importance. However, it remains a challenging task due to the complex spatial-temporal interactions among multiple observations. To this end, we select the lower-level (850 hPa) zonal daily wind component from region IMI2 (70E-90E, 20N-30N), provided on a spatial grid with a resolution of $1^0 \times 1^0$[38]. Wind speeds interact spatially and form a 231-dimensional subsystem. The FRMM performs reliable 20-day forward predictions for all observations, and the average $\rho$ and RMSE are 0.86 and 0.41, respectively (Fig. 4a–c and Supplementary Fig. S3). Additionally, the FRMM is certified by monthly observations in the same region (SI Appendix Fig. S4).

b) In neuroscience, electroencephalogram (EEG) signals have been extensively used to study the underlying mechanisms of the human brain as well as some typical diseases[39]. Therefore, ahead predictions of EEG signals are expected to deliver efficient early warnings for related diseases. EEG signals are often captured from different regions in the brain and show spatial-temporal dynamics. To test FRMM, we utilize a 64-dimensional subsystem that consists of EEG signal series from 64 channels from a healthy participant[40]. By setting the low-dimensional ($E = 5$) feature manifold of the system as a generalized predictor, we achieve accurate 20-second ahead predictions for all signals, where the average $\rho$ and RMSE are 0.92 and 0.42 (Fig. 4d–f and Supplementary Fig. S5).

c) Financial systems are typically complex systems influenced by numerous internal and external factors via various channels, resulting in high uncertainty and instability, which in turn makes prediction a difficult and challenging task[41]. We start with a 70-dimensional subsystem from the foreign exchange market, which includes the daily closing prices of 70 currencies against the US dollar. FRMM performs accurate 20-day-ahead predictions for all

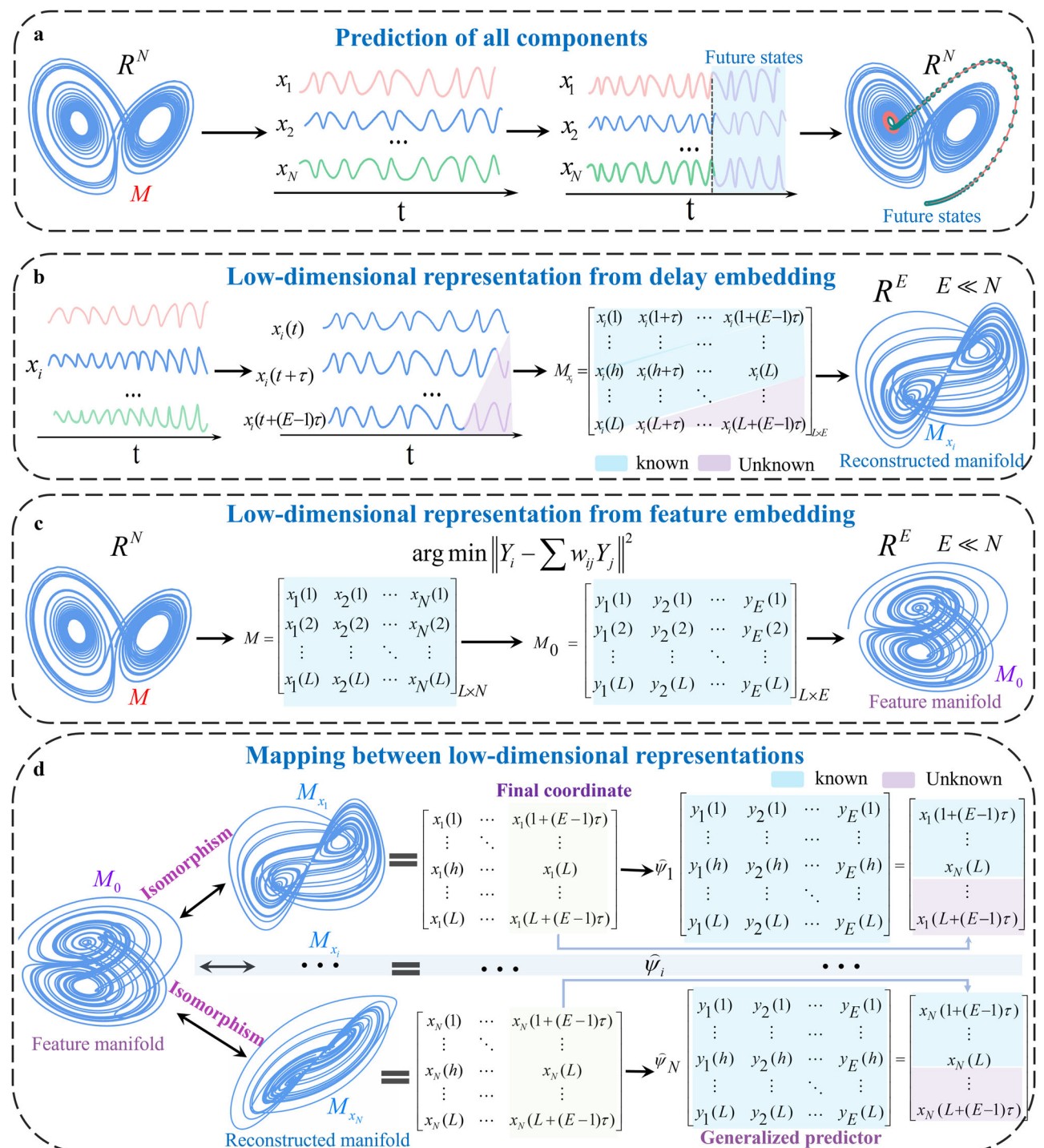

**Fig. 2 | Sketch of FRMM framework.** To forecast all components in an $N$-dimensional system (**a**), we find its $E$-dimensional representations from delay embedding (**b**) and feature embedding (**c**). Thus, an $N$-dimensional dynamical system $M$ is represented by two isomorphic low-dimensional manifolds (i.e., feature manifold $M_0$ and reconstructed manifold $M_{x_i}$). The foundation of an isomorphism suggests a one-to-one mapping between feature manifold $M_0$ and reconstructed manifold $M_{x_i}$. Then, it is possible to find a mapping $\widehat{\psi}_i(i=1,2,\cdots,N)$ from the feature manifold $M_0$ to the final coordinate of the reconstructed manifold $M_{x_i}$. Therefore, the feature manifold $M_0$ can be utilized as a generalized predictor to find the future dynamics (purple elements) of all components in complex systems (**d**).

observations, where the average $\rho$ and RMSE are 0.89 and 0.41 (Fig. 4g–i and Supplementary Fig. S6). In addition, the FRMM outputs reliable predictions of 46 stock indices from global stock markets; see SI Appendix Fig. S7.

d) Finally, we demonstrate the FRMM in a 207-dimensional traffic system, which consists of the traffic speeds collected from 207 loop detectors in Los Angeles County. Forecasting the time evolution of this traffic system is a challenging task due to the complex spatial and temporal dependencies among the elements of the system[42]. Our FRMM achieves 10-step ahead predictions ($\rho \geq 0.6$) for 86% (179) of the components, where the average $\rho$ and RMSE are 0.78 and 0.68 (the average accuracies for all components are 0.73 and 0.7) (Fig. 4j–l and Supplementary Fig. S8). As reported in Fig. 4l, the FRMM exhibits relatively poor performance for a few components, whose time series involve many abrupt changes, like tipping points, possibly caused by rush hours or accidents.

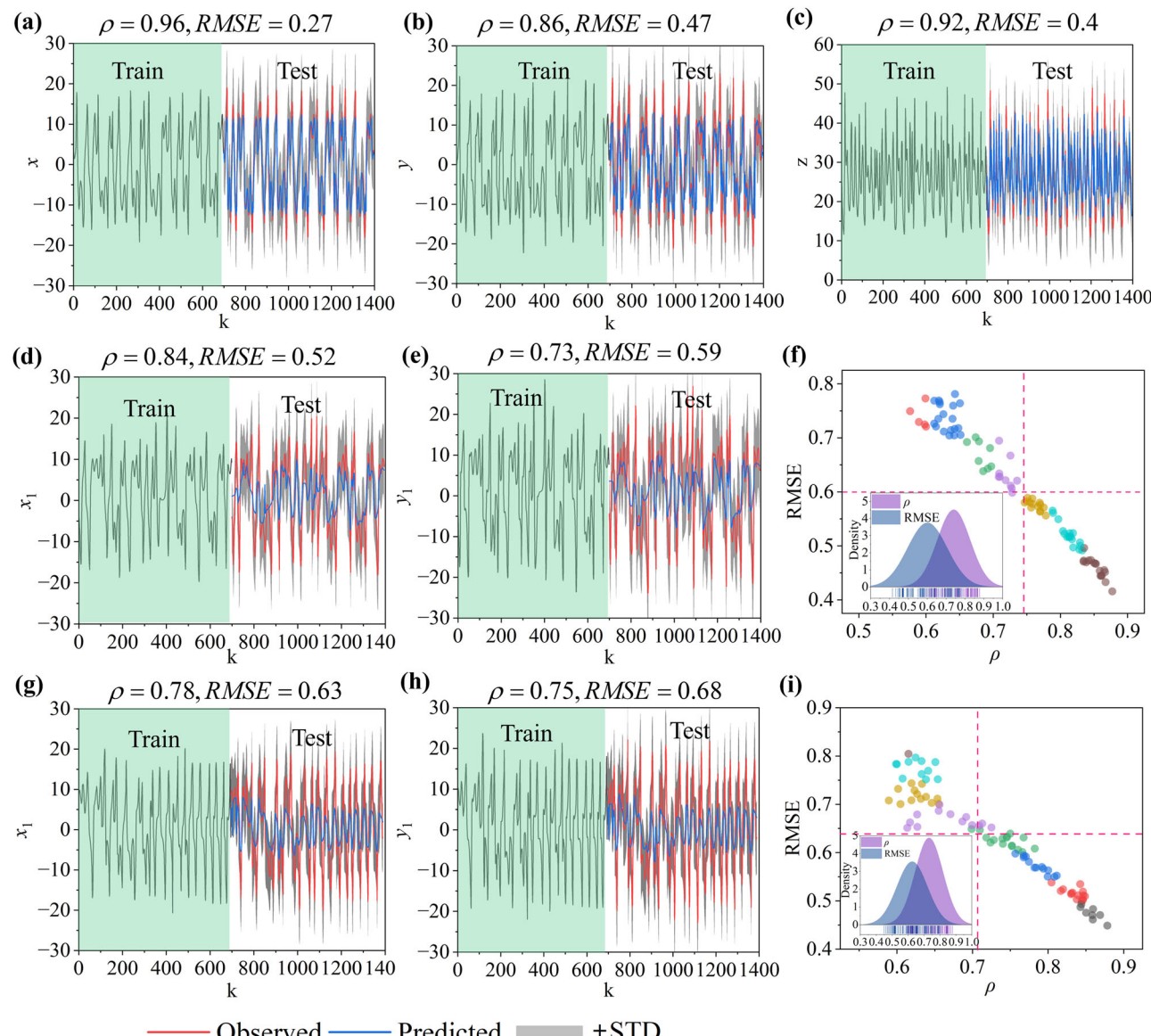

**Fig. 3 | Model systems. a–c** The performances of the 3-dimensional Lorenz system. **d–f** The performances of the 90-dimensional coupled Lorenz system, where the prediction accuracies of all 90 components are distributed in **f**, in which the red dotted line shows the average values of $\rho$ and RMSE. **g–i** The predictions of the 90-dimensional coupled Lorenz system with time-varying dynamics. The results demonstrate that the FRMM framework can output reliable multistep ahead predictions for all components in the Lorenz system. Note: We validate the accuracy by randomly selecting 50% of the series as a training sample (green shaded area), and the others are test samples. $k$ represents the randomly selected $k$-th data. STD represents the standard deviation of the observed series.

## Discussion

### Robustness tests

Generally, a predictive model performs better with longer training samples and shorter test samples, and the performance decreases sharply when training short samples but testing long samples. Our FRMM performs robustly even when inputting a short training sample (10% (140)) and verifying on a longer test sample (90% (1260)), where the average $\rho$ and RMSE are 0.69 and 0.75 (Fig. 5a, Supplementary Figs. S10 and S11). In addition, the FRMM is robust with deteriorating noise (Fig. 5b and Supplementary Fig. S12). For the length of input data, the FRMM also outputs reliable predictions with short input data (Fig. 5c). For the predicted step, the FRMM can find at most $(E − 1)\tau$ steps. Theoretically, each $\tau$ can be employed to reconstruct an isomorphic attractor if the observed series is long enough[30]. Often, only limited data are accessible, leading to poor reconstruction with an even larger $\tau$. Therefore, our framework is unable to make long-term predictions. Despite this, our framework achieves at most 30-step-ahead predictions for all components in the 90-dimensional ordinary Lorenz system (Fig. 5d).

### Remark on feature embedding

Identifying the low-dimensional feature manifold is critical for our prediction. However, embedding a false feature manifold (which has an inequivalent topology to the original system) may result in poor prediction due to the one-to-one mapping not being maintained between the feature manifold and the reconstructed manifold. For example, LLE and Laplacian fail to identify the 2-dimensional feature manifold of the 3-dimensional Lorenz system, resulting in poor predictions (variable z) by adding them to the FRMM framework (Fig. 6a, b). Conversely, despite the diffusion map algorithm, ISOMAP and LTSA also output a reliable feature manifold of the original attractor and could be used to perform accurate predictions for all variables by substituting the

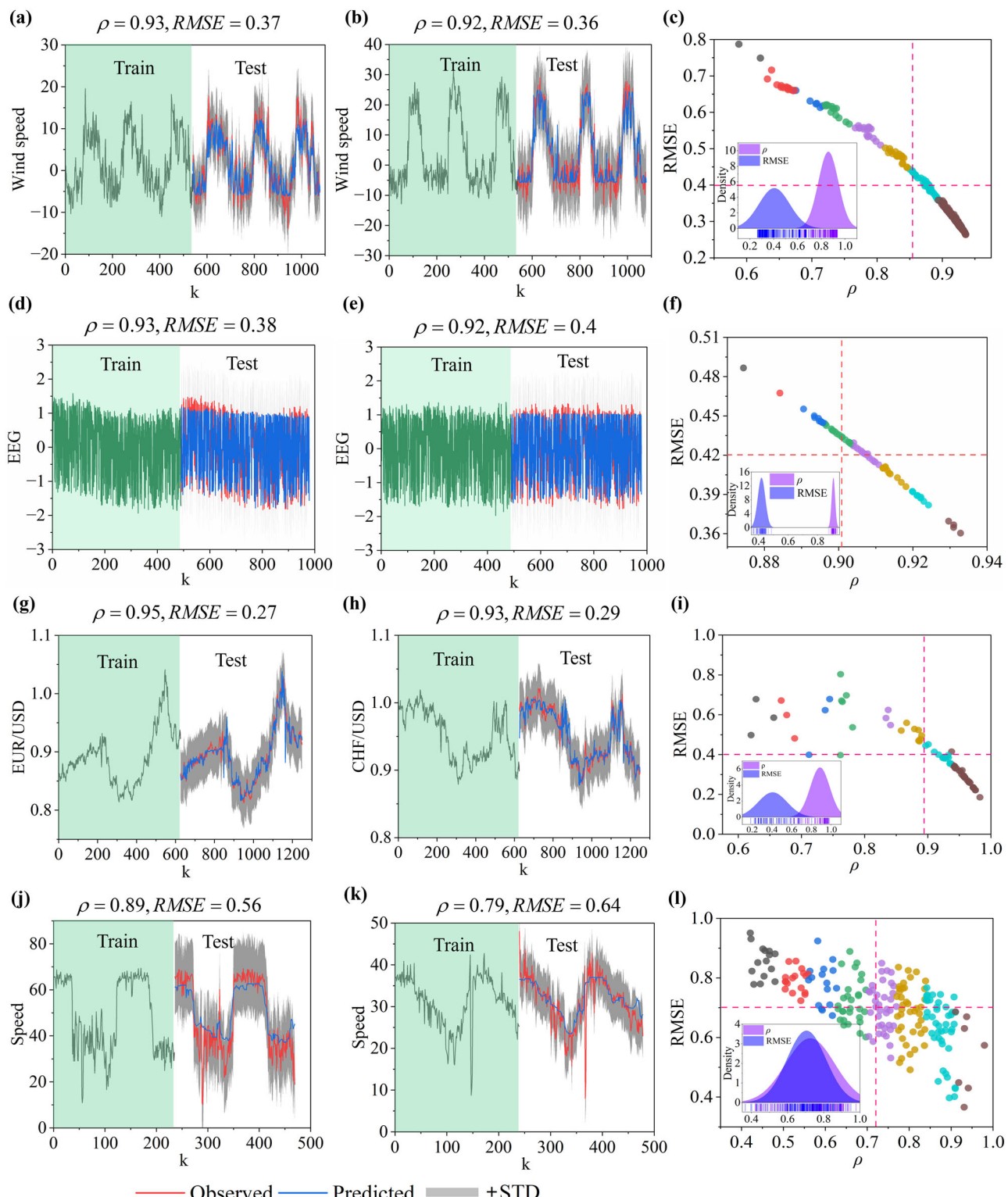

**Fig. 4 | Performance in real-world datasets.** The predictions of daily wind speed (m/s) (**a**–**c**), per second EEG signal (**d**–**f**), daily exchange rate (**g**–**i**), and traffic speed (5-minute interval) (**j**–**l**). By randomly selecting 50% of the data as a test sample, the FRMM is shown reliable for accurate multistep predictions in representative real-world systems, where the predicted horizons are $T = 20$ (**a**–**i**) and $T = 10$ (**j**–**l**).

diffusion map in the FRMM framework (Fig. 6c, d). However, the diffusion map outperforms ISOMAP and LTSA. Due to the diversity of dynamic structures in various real-world systems and the lack of sufficient details, only from their time series, there is no golden rule to select an optimal feature embedding algorithm (More discussions are given in SI Appendix Chapter 1.7).

Nevertheless, the five powerful techniques in this work make sense in many high-dimensional systems.

## Comparison with traditional methods
Many existing predictive models perform well when training on long samples and verifying on short samples, but the performances often

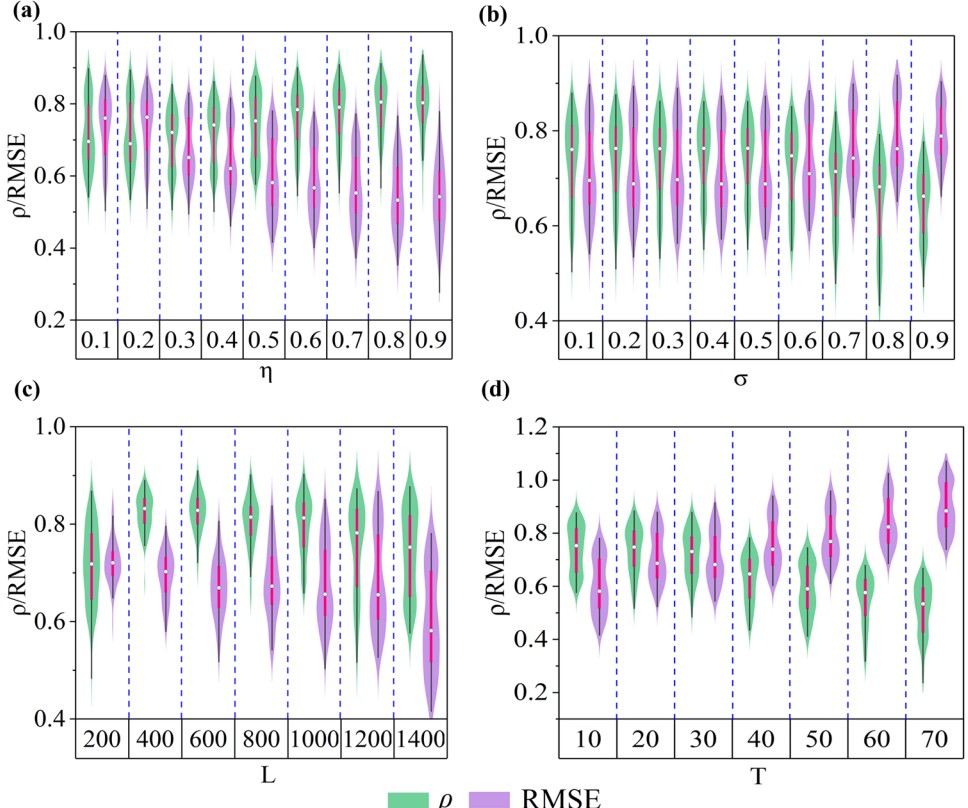

**Fig. 5 | Robustness tests.** We conduct tests of length of training sample (**a**), additive noise (**b**), length of input series (**c**), and predicted step (**d**). The 90-dimensional ordinary Lorenz system is used here, and the performance of all components is distributed as violins. FRMM performs better with longer training samples and shorter test samples, it remains reliable when training short samples but testing long samples (**a**). Given a target variable, FRMM can achieve multistep ahead predictions, but remains challenging for even longer horizons (**d**). Overall, the results demonstrate that the FRMM framework is robust for several fundamental factors. $\eta$ represents the proportion of randomly selected training samples. As with cross-validation, the longer the training sample is, the shorter the test sample. $\sigma$ represents the strength of additive white noise. $L$ is the length of the observed series. $T$ denotes the predicted step.

decrease sharply with short training samples and long test samples. We first compare the robustness of the length of training and test samples with some traditional methods (e.g., ARIMA[6], MVE[8], SVM[9], LSTM[10], RC[11]) and several advanced delay embedding-based frameworks (e.g., ARNN[15], RDE[14], MT-GPRM[13]. As depicted in Fig. 7, ARIMA fails to predict variable $z$ in the Lorenz system even when training on long samples, whereas the other methods predict it accurately. Although the performance decreases as the training sample length decreases and the test sample length increases, FRMM always remains relatively robust compared to other methods. Second, we compare the performance of our FRMM with the aforementioned methods across different datasets. The results indicate that FRMM yields relatively better predictions (Table 1). In summary, FRMM is shown more reliable for the predictions of all components in complex systems.

**Final remark on FRMM framework**
Our data-driven and model-free framework (FRMM) has been illustrated by both representative models and real-world systems, and it has several advantages. First, FRMM performs predictions by mapping between low-dimensional representations, which is well-grounded in theory that the topological structure of a high-dimensional dynamical system can be theoretically characterized in low-dimensional space from delay embedding and feature embedding. Second, for the uncertaties of the predictor and predictive model, FRMM sets the feature manifold as a generalized predictor to find future states of all components, and Gaussian process regression is utilized as a fixed tool to train all mappings between embedded manifolds. Third, many existing predictive

models directly train and fit relations between time series from a system, and they may perform poorly due to the inconstant correlations estimated from time series[43] (the fitted parameters in a model are often time-varying). Instead, FRMM finds the mapping between low-dimensional representations of a system, and this mapping is inherently supported. In summary, FRMM overcomes the curse of dimensionality, has higher interpretability, and shows potential to be applied in various fields.

Gaussian process regression is applied to find the mapping between the feature manifold and the reconstructed manifold. We need to note that this mapping can be also trained by some neural network algorithms, e.g., ARNN utilizes reservoir computing to train a mapping from the original attractor to the delay attractor. Despite the satisfactory performance of neural networks, they often rely on sufficient and rather large training samples. Besides, there remain unknown hidden details as black-box characters inside of some artificial neural networks. More importantly, the trade-offs between accuracy, cost, and interpretability are needed to be balanced in practical applications. On this basis, it seems more satisfactory to integrate Gaussian process regression in our FRMM framework.

FRMM is developed based on a popular framework, namely spatiotemporal information (STI) transformation[44]. Several advanced STI-based methods (e.g., MT-GPRM[13], RDE[14], and ARNN[15]) have been proposed to predict various complex systems. FRMM shows individual characteristics and meaningful improvements comparing with many existing STI-based methods. We clarify them from three aspects, including the prediction task, the architecture, and the theoretical foundation.

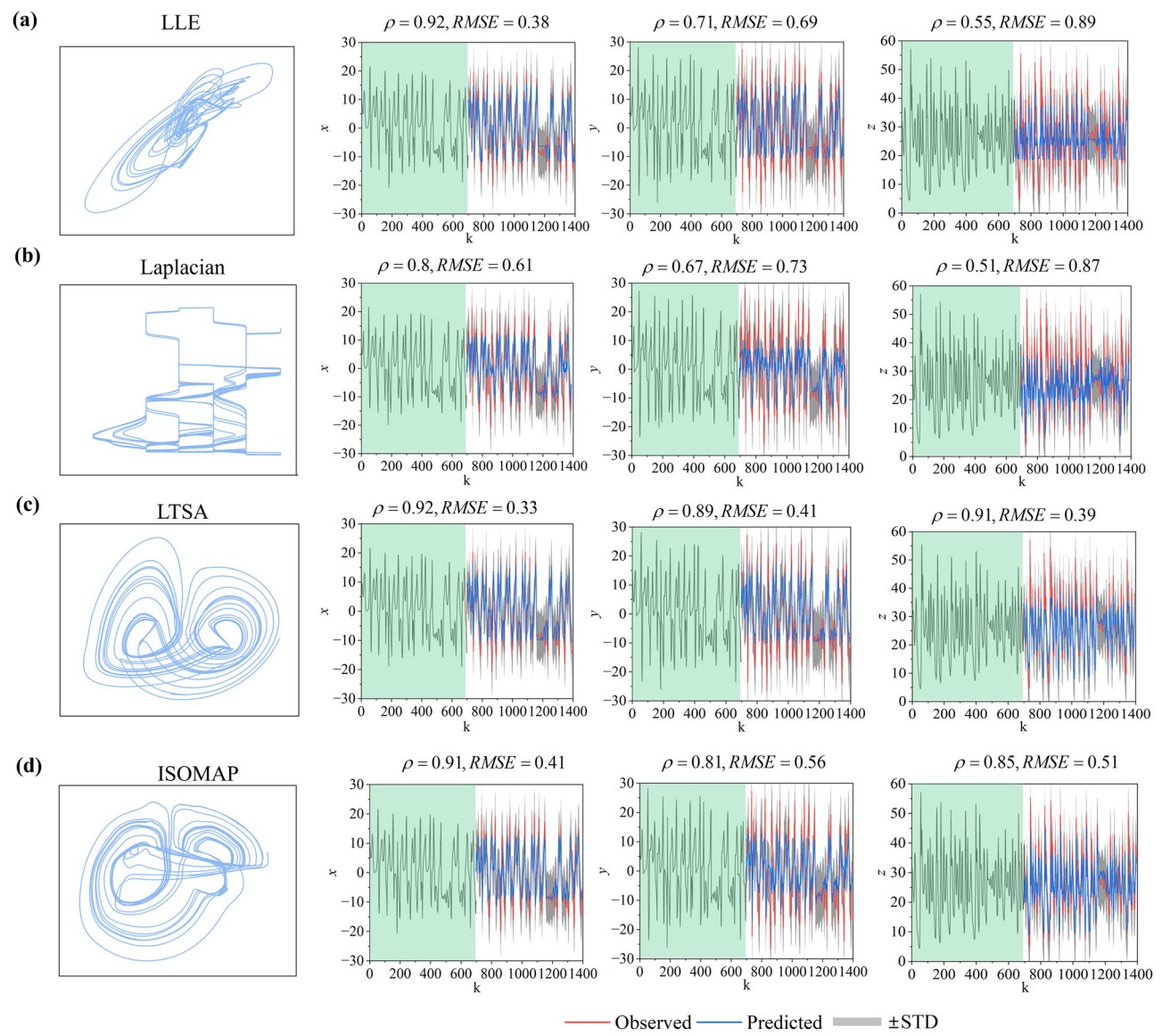

**Fig. 6 | Performance with other feature embedding techniques.** Several representative algorithms are considered, including LLE (**a**), Laplacian (**b**), ISOMAP (**c**), and LTSA (**d**). LLE and Laplacian algorithms fail to find faithful low-dimensional representations of Lorenz attractor, resulting in poor performance for some components (e.g., variable $z$). While LTSA and ISOMAP preserve the fundamental geometry of the original attractor, FRMM yields reliable predictions for all components. Several algorithms can be utilized for feature embedding in the 3-dimensional Lorenz system, and an integration of Diffusion map performs the best predictions among them (Fig. 3a).

For the prediction task, it is still unsolved for the predictions of all components in complex systems. Though some existing STI-based frameworks have the potential to address this issue, their abilities are often certified on partial components and not fully tested by all units in complex systems. Note that verifying a predictive model on fewer observations of complex systems may be risky. Take the generic 3-dimensional Lorenz system as an example (Eq. (16)), it is possible to predict variables $x$ and $y$ through a linear regression model, but this model fails to predict variable $z$[45]. It is uncritical to conclude that a linear regression model can predict the Lorenz system. In this direction, FRMM is faithful and exhibits higher potential for the predictions of all components in complex systems.

For the architecture, the main difference between FRMM and other STI-based frameworks is the selection of predictor. Some STI-based frameworks set the original system as the fixed predictor, e.g., MT-GPRM, while some frameworks may use different predictors for different targets, e.g., for each target variable, ARNN finds several

highly related components as predictors. FRMM focuses on system's fundamental dynamics and sets the system's low-dimensional feature manifold as a fixed and generalized predictor, which gives an efficient predictor when predicting different components in complex systems.

For the theoretical foundation, many existing STI-based frameworks create the STI equation by non-delay embedding and delay embedding, which originates from that a complex system can be approximately represented by different coordinates. Generally, the non-delay embedding of complex systems can be approximated in a space with either low or high dimension. (e.g., MT-GPRM sets all selected observations as a representation of original systems, RDE finds non-delay embedding by randomly selecting several observations). The theoretical foundation of FRMM is based on a well-accepted report that a high-dimensional system often has redundant information, and the system's fundamental dynamics (e.g., the topology of complex systems) are restored in low-dimensional manifolds[31–35]. FRMM framework

focuses on low-dimensional dynamics of complex systems, and these low-dimensional dynamics are identified by feature embedding and delay embedding. The feature embedding is conducted by powerful manifold learning algorithms, and these methods can automatically extract and restore the fundamental topology of the original system in a low-dimensional space. Thus, FRMM shows different theoretical foundations with existing STI-based frameworks. Additionally, identifying the fundamental dynamics of a high-dimensional system theoretically helps to reduce the negative impacts of redundant information in a high-dimensional system, and will be beneficial for better predictions. These are also supported by the relatively higher performance and robustness of FRMM in many real-world datasets (Table 1 and Fig. 7).

However, like other STI-based frameworks, FRMM fails to predict different components synchronously. In other words, for each target variable, one needs to train a suitable mapping. Additionally, FRMM also has limitations for situations in which a system experiences abrupt, rapid, and even irreversible transitions (known as tipping points)[46,47]. The behavior of a system shifts between contrasting states, and the historical rules are often not held when a system crosses the threshold, leading to poor predictions of our framework. The phenomenon of critical transitions, often caused by diverse external factors, is reported in numerous real-world systems. Despite this shortfall, the FRMM framework also inspires to identify the tipping points of a system, e.g., the occurrence of poor performance may indicate an underlying shift in the system.

## Methods

### FRMM framework
Given an $N$-dimensional system with time series $x_i(t)(i=1,2,\cdots,N,t=1,2,\cdots,L)$, we aim to predict all units within the system. For this task, the main structure of our FRMM framework is listed as follows:

(1) For each target variable $x_i(t)$, we estimate the embedding dimension $E$ (cf. SI Appendix Chapter 3.3) and time lag $\tau$ based on the false nearest neighbor algorithm and mutual information function, respectively[48,49]. Then, this approach allows for reconstructing an isomorphic manifold $M_{x_i}$ in an $E$-dimensional space ($E$ is usually much smaller than $N$). The reconstructed manifold $M_{x_i}$ is given as

$$M_{x_i} = \begin{bmatrix} x_i(1) & x_i(1+\tau) & \cdots & x_i(1+(E-1)\tau) \\ \vdots & \vdots & \cdots & \vdots \\ x_i(h) & x_i(h+\tau) & \cdots & x_i(L) \\ \vdots & \vdots & \cdots & \vdots \\ x_i(L) & x_i(L+\tau) & \cdots & x_i(L+(E-1)\tau) \end{bmatrix}, \quad (5)$$

where $L$ is the length of the series and $h=L-(E-1)\tau$. Note that the elements from $x_i(1)$ to $x_i(L)$ are observed values from the original system, others ($x_i(L+\tau),\cdots,x_i(L+(E-1)\tau)$) are unknown, and our goal is to predict them. According to Takens, each time series variable can be used to reconstruct an $E$-dimensional manifold[30], which gives the fundamental basis for predicting all components in complex systems.

(2) Moreover, the low-dimensional manifolds for the system can also be identified by preserving their fundamental geometric features (feature embedding). In this work, we provide several techniques to find low-dimensional representations for the systems, e.g., isometric feature mapping (ISOMAP), locally linear embedding (LLE), Laplacian, diffusion map, and local tangent space alignment (LTSA), since real-world systems often behave with different dynamical structures and have various geometric features.

We select LLE as an example to clarify the main idea of low-dimensional embedding. Given an $N$-dimensional dimensional system with observed vectors $X_i=(x_{1i},x_{2i},\cdots,x_{Ni})$, we approximate each point

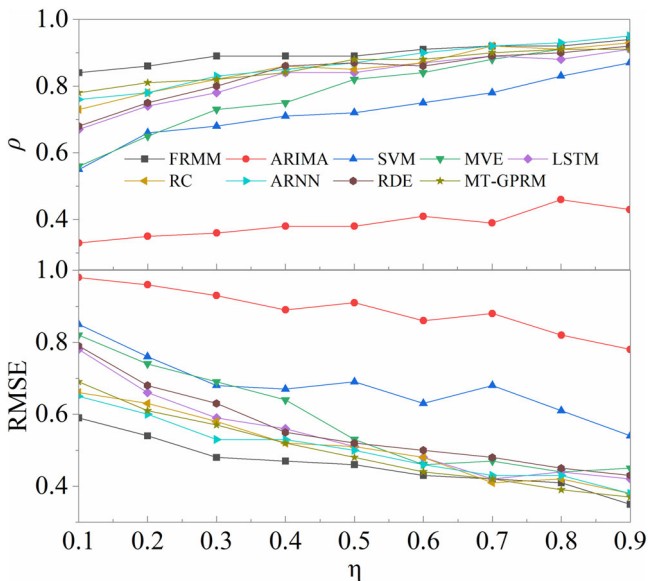

**Fig. 7 | Comparison of robustness with classic predictive models concerning the length of the training sample, including ARIMA, SVM, MVE, LSTM, RC, ARNN, RDE, and MT-GPRM.** We compare the robustness of the one-step ahead prediction of variable $z$ from the 3-dimensional Lorenz system. Many methods show reliable predictions when inputting long training samples, while our FRMM remains robust when training on short samples and verifying on long samples. $\eta$ gives the proportion of the training sample.

**Table 1 | Comparison of performance with several classic approaches**

| Real-world dataset | Metric | Method | | | | | | | | |
|---|---|---|---|---|---|---|---|---|---|---|
| | | FRMM | ARNN | RDE | MVE | SVM | LTSM | RC | MT-GPRM | ARIMA |
| Wind speed | $\rho$ | 0.86 | 0.88 | 0.61 | 0.52 | 0.45 | 0.61 | 0.67 | 0.83 | 0.12 |
| | RMSE | 0.41 | 0.51 | 0.79 | 0.97 | 0.88 | 0.85 | 0.7 | 0.52 | 0.89 |
| EEG signal | $\rho$ | 0.89 | 0.68 | 0.51 | 0.45 | 0.34 | 0.38 | 0.56 | 0.61 | 0.05 |
| | RMSE | 0.6 | 0.74 | 0.79 | 1.03 | 0.91 | 0.82 | 0.89 | 0.87 | 1.07 |
| Exchange rate | $\rho$ | 0.89 | 0.81 | 0.66 | 0.61 | 0.42 | 0.51 | 0.67 | 0.84 | 0.23 |
| | RMSE | 0.41 | 0.63 | 0.85 | 0.89 | 0.8 | 0.92 | 0.84 | 0.44 | 0.98 |
| Traffic speed | $\rho$ | 0.73 | 0.67 | 0.56 | 0.42 | 0.17 | 0.31 | 0.53 | 0.7 | -0.18 |
| | RMSE | 0.7 | 0.82 | 0.91 | 1.01 | 0.94 | 0.97 | 0.96 | 0.68 | 1.02 |

Note: We compare the average performance for all components in these systems, where 50% of the observed series are used as test samples (cross-validation). Two accuracy metrics are employed, including $\rho$ (Pearson correlation between predicted values and observed values, see the number in the first row for each dataset) and RMSE (Normalized by the standard deviation of the input series, see the number in the second row for each dataset). All the datasets are the same to those in the main experiments in *Results*. All simulations are operated in Matlab 2018a, with the exception of MVE prediction, which is conducted using package "rEDM", in R.

by a linear function of its $K$ nearest neighbors (e.g., $K = 8$).

$$\tilde{X}_i = \sum_{i=1}^{N} w_{ij} X_j, \sum_j w_{ij} = 1, \qquad (6)$$

where $w_{ij}$ measures the weight between the $i$th point and $j$th point. To find the optimal set of weights $\widehat{W} = (\widehat{w}_{ij})$, we minimize the loss function

$$\widehat{W} = \arg \min \sum_{i=1}^{N} \left\| X_i - \sum_{j=1}^{N} w_{ij} X_j \right\|^2. \qquad (7)$$

We expect the local geometry in the original space to be preserved in their low-dimensional manifold. Therefore, we fix the matrix $\widehat{W} = (\widehat{w}_{ij})$ and find the low-dimensional embedding by solving

$$\widehat{Y} = \arg \min_Y \sum_{i=1}^{N} \left\| Y_i - \sum_{j=1}^{N} \widehat{w}_{ij} Y_j \right\|^2, \qquad (8)$$

where $Y_i$ represents the points in the low-dimensional manifold. Then, the bottom $E$ nonzero eigenvectors (from Eq. (8)) provide the low-dimensional embedding $M_0$

$$M_0 = \begin{bmatrix} y_1(1) & \cdots & y_E(1) \\ \vdots & \ddots & \vdots \\ y_1(L) & \cdots & y_E(L) \end{bmatrix}, \qquad (9)$$

where $y(t) \in Y_i$. Consequently, each $N$-dimensional observation $X_i$ is mapped to an $E$-dimensional point $Y_i$.

(3) An $N$-dimensional system is embedded into an $E$-dimensional space from two different approaches, which then suggests a one-to-one mapping between $M_{x_i}$ and $M_0$.

$$\psi_i : M_0 \to M_{x_i}, \psi_i(Y(t)) = \tilde{X}_i(t), i = 1, 2, \cdots, N, \qquad (10)$$

where $Y(t) \in M_0, \tilde{X}_i(t) \in M_{x_i}$. From Eq. (10), we infer

$$\psi_i \begin{pmatrix} y_1(1) & y_2(1) & \cdots & y_E(1) \\ \vdots & \vdots & \cdots & \vdots \\ y_1(h) & y_2(h) & \cdots & y_E(h) \\ \vdots & \vdots & \cdots & \vdots \\ y_1(L) & y_2(L) & \cdots & y_E(L) \end{pmatrix} = \begin{pmatrix} x_i(1) & x_i(1+\tau) & \cdots x_i(1+(E-1)\tau) \\ \vdots & \vdots & \cdots & \vdots \\ x_i(h) & x_i(h+\tau) & \cdots & x_i(L) \\ \vdots & \vdots & \cdots & \vdots \\ x_i(L) & x_i(L+\tau) & \cdots x_i(L+(E-1)\tau) \end{pmatrix}. \qquad (11)$$

Since a one-to-one mapping between the feature manifold and the reconstructed manifold is held, it is possible to train a mapping from the feature manifold to each coordinate of the reconstructed manifold. In particular, for longer horizon predictions, we aim to find the mapping from the feature manifold to the final coordinate of the reconstructed manifold (i.e., $x(t + (E-1)\tau), t = 1, 2, \cdots, L$). These processes can be also explained mathematically, as follows.

Based on Eq. (11), we deduce the form (12)

$$\widehat{\phi}_i \begin{pmatrix} x_i(1) & x_i(1+\tau) & \cdots x_i(1+(E-1)\tau) \\ \vdots & \vdots & \cdots & \vdots \\ x_i(h) & x_i(h+\tau) & \cdots & x_i(L) \\ \vdots & \vdots & \cdots & \vdots \\ x_i(L) & x_i(L+\tau) & \cdots x_i(L+(E-1)\tau) \end{pmatrix} = \begin{pmatrix} x_i(1+(E-1)\tau) \\ \vdots \\ x_i(L) \\ \vdots \\ x_i(L+(E-1)\tau) \end{pmatrix}. \qquad (12)$$

In Eq. (12), $\widehat{\phi}_i$ can be easily obtained by the transform (13)

$$\begin{pmatrix} x_i(1) & x_i(1+\tau) & \cdots x_i(1+(E-1)\tau) \\ \vdots & \vdots & \cdots & \vdots \\ x_i(h) & x_i(h+\tau) & \cdots & x_i(L) \\ \vdots & \vdots & \cdots & \vdots \\ x_i(L) & x_i(L+\tau) & \cdots x_i(L+(E-1)\tau) \end{pmatrix} \begin{pmatrix} 0 \\ \vdots \\ [0] \\ \vdots \\ 1 \end{pmatrix} = \begin{pmatrix} x_i(1+(E-1)\tau) \\ \vdots \\ x_i(L) \\ \vdots \\ x_i(L+(E-1)\tau) \end{pmatrix}. \qquad (13)$$

According to Eqs. (11) and (12), we deduce a form (14)

$$\widehat{\psi}_i \begin{pmatrix} y_1(1) & y_2(1) & \cdots & y_E(1) \\ \vdots & \vdots & \cdots & \vdots \\ y_1(h) & y_2(h) & \cdots & y_E(h) \\ \vdots & \vdots & \cdots & \vdots \\ y_1(L) & y_2(L) & \cdots & y_E(L) \end{pmatrix} = \begin{pmatrix} x_i(1+(E-1)\tau) \\ \vdots \\ x_i(L) \\ \vdots \\ x_i(L+(E-1)\tau) \end{pmatrix}, \qquad (14)$$

where $\widehat{\psi}_i(x) = \widehat{\phi}_i \psi_i(x)$. Equation (14) suggests a mapping from the feature manifold to the final coordinate of the reconstructed manifold.

All the elements on $M_0$ (see the left matrix in Eq. (14)) are obtained via manifold learning algorithms, whereas partial components on $M_{x_i}$ are unknown, i.e., $x_i(L+\tau), \cdots, x_i(L+(E-1)\tau)$ (see the right matrix in Eq. (14)), and they represent the future dynamics of the variable $x_i(t)$. Once $\psi_i$ is identified, one can find at most $(E-1)\tau$-forward dynamics of a selected variable $x_i(t)$.

In a dynamical system, each time series variable can be used to reconstruct a low-dimensional embedding (i.e., $M_{x_i}$, where $i = 1, 2, \cdots, N$). This approach thus enables the construction of $N$ mappings (i.e., $\psi_i, i = 1, 2, \cdots, N$) from $M_0$ to the final coordinate of $M_{x_i}$, which yields multistep predictions for all units in high-dimensional dynamical systems. In this work, we use the Gaussian process regression algorithm to identify every $\widehat{\psi}_i$.

(4) We validate the performance by randomly dividing the observed series $(x_i(1+(E-1)\tau), x_i(2+(E-1)\tau), \cdots, x_i(L))$ into a training set and a test set (i.e., cross-validation). The correlation between the observed values and predicted values ($\rho$) and the normalized root mean square error (RMSE) are applied to measure the performance.

$$\rho = \frac{\mathrm{cov}(x, \tilde{x})}{\eta_x \eta_{\tilde{x}}}, RMSE = \frac{\sqrt{\frac{1}{n} \sum (x - \tilde{x})^2}}{\eta_x}, \qquad (15)$$

where $x$ and $\tilde{x}$ are the original and predicted data, respectively. $\eta_x$ represents the standard deviation of the series $x$.

## Benchmark model systems

The coupled Lorenz system is defined as Eq. (16), where the $i$th ($i = 1, 2, \cdots, N$) subsystem is coupled with the $(i-1)$ subsystem via $c$. To make the system closed, we set $i-1$ as $N$ for $i = 1$.

$$\begin{aligned} \dot{x}_i &= \sigma(t)(y_i - x_i) + cx_{i-1}, \\ \dot{y}_i &= ax_i - y_i - x_i z_i, \\ \dot{z}_i &= bz_i + x_i y_i, \end{aligned} \qquad (16)$$

where $\sigma(t)$ is the time-varying parameter. $a$ and $b$ are set to be generic values, i.e., $a = 28, b = -8/3$.

For the time-invariant case ($\sigma(t) \equiv 10$), Eq. (16) depicts an ordinary Lorenz system. Particularly, we obtain a 3-dimensional Lorenz system when $N = 1$ and $c = 0$. We define a 90-dimensional coupled Lorenz system when $N = 30$ and $c = 0.1$.

For the time-varying case, we set $\sigma(t)$ to be increased (from an initial value of 10) by 0.2 after every ten time intervals, i.e., $\sigma(t) = 10 + 0.2(t|10)$.

To generate the discrete data, we set the initial state as 0.1, and the output series has a length of 1500. The first 100 data points are ignored to avoid transient dynamics. We embed the 3-dimensional ordinary Lorenz system into a 2-dimensional space, where $E = 2, \tau = 10$. For the 90-dimensional ordinary and time-varying Lorenz systems, the embedding dimension and time lag are $E = 11$ and $\tau = 1$, respectively. The diffusion map algorithm is used to find their low-dimensional representations.

## Reporting summary

Further information on research design is available in the Nature Portfolio Reporting Summary linked to this article.

## Data availability

The details of real-world datasets are listed in SI Appendix Chapter 1.2. All real-world datasets are available at https://github.com/wt1234wt/FRMM-framework. The details of datasets from model systems are given in Methods.

## Code availability

The related codes are available at https://github.com/wt1234wt/FRMM-framework.

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

## Acknowledgements

X.Y.G. is supported by the National Natural Science Foundation of China (No.72371229, 71991481, 71991485), and by the Fundamental Research Funds for the Central Universities (3-7-8-2023-01). H.Z.A. is supported by the National Natural Science Foundation of China (No.71991481, 71991480], and by the Basic Science Center Project of the National Natural Science Foundation of China (No. 72088101). J.X.G. is supported by the National Science Foundation (No. 2047488), and by the Rensselaer- IBM AI Research Collaboration.

## Author contributions

T.W., X.Y.G., F.A., J.X.G. and J.K. designed the research. T.W., X.Y.G. and F.A. performed the experiments and wrote the first draft of the paper. T.W., X.T.S., S.G. and Z.S. analyzed data. T.W., X.Y.G., F.A., H.Z.A., J.X.G. and J.K. reviewed and edited the manuscript.

## Competing interests

The authors declare no competing interests.
