## [Peer Review File · Nature Communications]

REVIEWER COMMENTS

Reviewer #1 (Remarks to the Author):

The author proposes a data-driven prediction method based on feature embedding and delay embedding, aiming to predict all Networked Components in Complex Systems with accuracy and efficiency for long-term forecasting. To some degree, the paper is interesting, but still needs to be improved.

some comments:

1. The article utilizes the Gaussian process regression algorithm to solve ψ_i , which shows some similarity to the GPRM method (10.1016/j.ins.2022.11.159). Please provide a more detailed explanation of the similarities and differences between GPRM and FRMM. Can ψ_i be solved using neural networks?
2. In the introduction section, the author mentions methods like RDE, ARNN, which are also based on delay embedding. However, in Fig. 7, there is no comparison with any delay embedding-based methods. I suggest conducting experiments and discussions for each data type with different methods. Particularly, what are the average effects across datasets? In what scenarios does FRMM perform well or poorly? Use a table to present the results.
3. In the introduction section, the author emphasizes the necessity of predicting all networked components in a complex system and underscores "computational cost". How does FRMM achieve computational cost efficiency compared to other methods? For each x_i that needs prediction, a function ψ_i must be trained. What is the fundamental difference compared to the approach "regarding all the remaining variables as predictors"? Please elaborate further based on comment 2.
4. The author claims that FRMM can "predict all network components," which is the most confusing point for me. I initially thought the author was going to establish a network in advance, but neither the manuscript nor SI explicitly defines what the network is.
5. The paper combines delay embedding and feature embedding to propose FRMM. Can a dissection experiment be designed to confirm the necessity of FRMM components?

6. It is suggested to integrate Fig. S1 with Fig. 2 and provide a clearer FRMM framework in the main text. Besides, what do the different colors of points in Fig. 1 represent?

7. The Takens' embedding theory is usually used in a steady state, but the EGG data is non-stationary, how does the author predict non-stationary EGG data? Is preprocessing needed? Is denoising necessary?

8. Concerning different data, is there a criterion for choosing which dimension reduction method for feature embedding to use? How did the author determine which method to adopt?

Reviewer #2 (Remarks to the Author):

Authors proposed a data-driven and model-free framework, FRMM, by combining feature and delay embedding to predict all components of a complex networks. It is claimed that this is a universal predictor and can be applied to high to low dimensional dynamical systems and verified by real data. In the last two decades, it was an open problem to design a generalized framework for the predictions of all networked components in a complex system.

The results are interesting and demand for publication after a minor editing:

1. In delay embedding, is there any methods to choose delay τ ? In result section, $\tau=10$ was chosen. Also the choice of the parameter E .
2. In Fig. 1, the axes labels are missing.
3. In Eq. (14), why the parameter $b=-8/3$? In Lorenz system the parameter b is positive, I think this is a typo mistake. Please check.
4. In real systems, one cannot avoid the presence of noise. Is the proposed method applied for the presence of small noise? Discussion on this is needed.
5. The figure caption should be in details, at least the finding of main results.
6. Authors verified the proposed techniques for two model simulations and 4 real data. It is claimed that the predictor is universal. Why is it universal? Will it be applied for other systems and other network topologies?

Overall, I have enjoyed the reading of the results and the results are really interesting. Before publication, the above points may be discussed.

Reviewer #1 (Remarks to the Author):

The author proposes a data-driven prediction method based on feature embedding and delay embedding, aiming to predict all Networked Components in Complex Systems with accuracy and efficiency for long-term forecasting. To some degree, the paper is interesting, but still needs to be improved.

Response: Thanks for your constructive comments and suggestions, which help to improve our work greatly. The point-to-point responses are given as follows. Hope that we have addressed all your concerns.

Note that Fig. SS1, Fig. SS2, and Fig. SS3 are used to explain related issues in this letter and not be contained in our article. The modified contents are noted by blue color, see attached *Revised manuscript with notes* and *Revised SI with notes*. Also, we give clear main manuscript and SI.

some comments:

1. The article utilizes the Gaussian process regression algorithm to solve ψ_i , which shows some similarity to the GPRM method (10.1016/j.ins.2022.11.159). Please provide a more detailed explanation of the similarities and differences between GPRM and FRMM. Can ψ_i be solved using neural networks?

Response: Thanks for your comments. The mentioned work proposed a new reliable framework for the prediction task, namely multitask learning-based Gaussian process regression machine (MT-GPRM). This novel framework shows some similar characteristics to our FRMM, but still, some individual aspects exist between them.

Similarity:

i) Both MT-GPRM and FRMM are data-driven approaches and achieve ahead predictions by mapping spatial state points to a temporal series, namely a spatiotemporal information transformation (STI). Specifically, the STI equation can be given as $\psi(X(t)) = Y(t)$, where $X(t)$ is the selected predictor and $Y(t)$ is often constructed based on delay embedding from a target time series variable.

ii) Both Gaussian process regression and delay embedding are utilized to construct the prediction framework. Gaussian process regression is used to solve the needed mapping. Delay embedding helps to construct a well-grounded foundation for the prediction, which allows for the reconstruction of an isomorphic attractor with the original system from a single time series variable, and this variable is the goal to be predicted.

Difference:

i) The main difference between MT-GPRM and FRMM is the predictor. MT-GPRM sets all system components as predictors, while FRMM identifies a low-dimensional feature manifold (feature embedding) as a generalized predictor.

ii) Their operational rules are different. The main framework of MT-GPRM is set as (1)

$$\begin{pmatrix} \psi_1(X(t_1)) & \psi_1(X(t_2)) & \cdots & \psi_1(X(t_L)) \\ \psi_2(X(t_1)) & \psi_2(X(t_2)) & \cdots & \psi_2(X(t_L)) \\ \cdots & \cdots & \ddots & \cdots \\ \psi_L(X(t_1)) & \psi_L(X(t_2)) & \cdots & \psi_L(X(t_L)) \end{pmatrix} = \begin{pmatrix} x_i(t_1) & x_i(t_2) & \cdots & x_i(t_L) \\ x_i(t_2) & x_i(t_3) & \cdots & x_i(t_{L+1}) \\ \cdots & \cdots & \ddots & \cdots \\ x_i(t_L) & x_i(t_{L+1}) & \cdots & x_i(t_{L+L-1}) \end{pmatrix}, \quad (1)$$

where $X(t_i) \in R^N$, L is the length of the time series, $x_i(t)$ is a target time series variable from the original system. The right matrix is constructed by target time series based on delay embedding, where the embedding dimension and time lag are fixed for all datasets, i.e., $E = L$ and $\tau = 1$. The elements from $x_i(t_1)$ to $x_i(t_L)$ are known before, while the elements from $x_i(t_{L+1})$ to $x_i(t_{L+L-1})$ are unknown and needed to be predicted. Then, MT-GPRM conducts prediction by the average performance of multitask Gaussian process regression. For example, if we want to find $x_i(t_{L+1})$, there are $L-1$ available mappings, i.e.,

$$\psi_2(X(t_L)) = \psi_3(X(t_{L-1})) = \cdots = \psi_L(X(t_2)) = x_i(t_{L+1}) \quad (2)$$

The final prediction of $\hat{x}_i(t_{L+1})$ can be estimated by the average predictions from $L-1$ mappings (Eq.3).

$$\hat{x}_i(t_{L+1}) = \frac{1}{L-1} \sum_{i=2}^L \psi_i(X(t_{L-i+2})) \quad (3)$$

Theoretically, MT-GPRM can output at most $(L-1)$ step ahead predictions. However, the practical predicted step would be shorter because there is less training sample to find reliable mapping ψ_i when i approaches L , e.g., there is only one sample to train the last mapping ψ_L , that is $\psi_L(X(t_L)) = x_i(t_{L+L-1})$. Moreover, an isomorphic reconstruction is needed to guarantee the existence of a one-to-one mapping from the original attractor to its delay attractor. MT-GPRM sets fixed parameters for delay embedding (i.e., $E = L$ and $\tau = 1$), and this may cause undesirable reconstruction in some conditions, e.g., the reconstructed attractor may not unfold when the time lag is even small (Ref. 45 in main text), e.g., $\tau = 1$. To guarantee an isomorphic reconstruction with the original system, there are several criteria for the selection of embedding dimension and time lag, e.g., false nearest neighbor and mutual information.

The main framework of FRMM is given as

$$\psi \begin{pmatrix} y_1(1) & y_2(1) & \cdots & y_E(1) \\ \vdots & \vdots & \cdots & \vdots \\ y_1(h) & y_2(h) & \cdots & y_E(h) \\ \vdots & \vdots & \cdots & \vdots \\ y_1(L) & y_2(L) & \cdots & y_E(L) \end{pmatrix} = \begin{pmatrix} x_i(1) & x_i(1+\tau) & \cdots & x_i(1+(E-1)\tau) \\ \vdots & \vdots & \cdots & \vdots \\ x_i(h) & x_i(h+\tau) & \cdots & x_i(L) \\ \vdots & \vdots & \cdots & \vdots \\ x_i(L) & x_i(L+\tau) & \cdots & x_i(L+(E-1)\tau) \end{pmatrix}, \quad (4)$$

where the left matrix is identified by feature embedding and the right matrix is obtained by delay embedding from a single time series. Considering the final column of the right matrix in Eq. (3), the elements from $x_i(1+(E-1)\tau)$ to $x_i(L)$ are known and can be used to train the mapping ψ , while the elements from $x_i(L+1)$ to $x_i(L+(E-1)\tau)$ are unknown and needed to be predicted. Once finding the ψ , FRMM yields at most $(E-1)\tau$ step ahead predictions of $x_i(t)$.

For each target variable, MT-GPRM achieves ahead predictions by the average performance from multitask Gaussian process regression, while FRMM addresses prediction via a single Gaussian process regression. FRMM has different operational rules compared with MT-GPRM.

iii) The computational cost is different. For a target variable, MT-GPRM sets all system components as a predictor and outputs the final prediction by the average performance from J mappings ($J \in \{1, 2, \dots, L-1\}$). For each mapping, the computational cost is approximately $O(N^3)$ (N is the input dimension). Thus, the total computational cost of MT-GPRM is approximately $O(JN^3)$. The cost of FRMM is less than $O(NL \log(L) + L^2(2 + \log(L)))$ (see details in Comment 3). Generally, MT-GPRM would take a relatively higher cost for high-dimensional inputs but short series.

In summary, both MT-GPRM and FRMM achieve predictions by constructing a spatiotemporal information transformation. However, their operational rules are quite different. FRMM shows higher performance than MT-GPRM (see comment 2). FRMM extends the STI equation by an innovative combination of feature embedding and delay embedding, which exhibits great potential for reliable predictions of all components in complex systems.

Gaussian process regression is applied to find the mapping between the feature manifold and the reconstructed manifold. We need to note that this mapping can be also trained by some neural network algorithms, e.g., ARNN utilizes reservoir computing to train a mapping from the original attractor to the delay attractor. Despite the satisfactory performance of neural networks, they often rely on sufficient and rather large training samples. Besides, there remain unknown hidden details as black-box characters inside

of some artificial neural networks. More importantly, the trade-offs between accuracy, cost, and interpretability are needed to be balanced in practical applications. On this basis, it seems more satisfactory to integrate Gaussian process regression in our FRMM framework.

We add comparisons with MT-GPRM, see Table 1 and Fig. 7 in the main text. Then, we add the discussions of neural network to solve ψ in Discussion in the main text (Line 370-379).

2. In the introduction section, the author mentions methods like RDE, ARNN, which are also based on delay embedding. However, in Fig. 7, there is no comparison with any delay embedding-based methods. I suggest conducting experiments and discussions for each data type with different methods. Particularly, what are the average effects across datasets? In what scenarios does FRMM perform well or poorly? Use a table to present the results.

Response: Thanks for your suggestions. As you mentioned, RDE and ARNN are two state-of-the-art delay embedding-based methods, we add experiments to compare the performance for each data type with different methods. As we know, both non-delay embedding (e.g., finding a low-dimensional representation by randomly selecting E variables from an N -dimensional system) and delay embedding can find isomorphic and low-dimensional representations of original systems. On this base, RDE aims to find the mapping from the non-delay attractor (non-delay embedding) to the delay attractor (delay embedding). For an N -dimensional system, there may output $q = C_N^E$ non-delay embeddings. Thus, one needs to train q mappings from q non-delay attractors to delay attractors. Then, there may output q predictions for each target variable. The final prediction can be estimated by an expectation process. ARNN is a combination of delay embedding and a neural network algorithm (i.e., reservoir computing), which addresses ahead predictions by training a mapping from the original manifold to a delay manifold.

We add comparisons with several classic methods with different datasets, see Table 1. FRMM has higher average performance for the predictions for all components across different datasets.

Table. 1 Comparison with several classic approaches

Dataset	Method								
	FRMM	ARNN	RDE	MVE	SVM	LTSM	RC	MT-GPRM	ARIMA

Wind speed	0.86	0.88	0.61	0.52	0.45	0.61	0.67	0.83	0.12
	0.41	0.51	0.79	0.97	0.88	0.85	0.7	0.52	0.89
EEG signal	0.89	0.68	0.51	0.45	0.34	0.38	0.56	0.61	0.05
	0.6	0.74	0.79	1.03	0.91	0.82	0.89	0.87	1.07
Exchange rate	0.89	0.81	0.66	0.61	0.42	0.51	0.67	0.84	0.23
	0.41	0.63	0.85	0.89	0.8	0.92	0.84	0.44	0.98
Traffic speed	0.73	0.67	0.56	0.42	0.17	0.31	0.53	0.7	-0.18
	0.7	0.82	0.91	1.01	0.94	0.97	0.96	0.68	1.02

Note: We compare the average performance for all components in these systems, where 50% of the observed series are used as test samples (cross-validation). Two accuracy metrics are employed, including ρ (Pearson correlation between predicted values and observed values, see first element in the box) and RMSE (Normalized by the standard deviation of the input series). All the datasets are the same to those in the main experiments in *Results*. All simulations are operated in Matlab 2018a, with the exception of MVE prediction, which is conducted using package “rEDM”, in R.

Moreover, we add comparisons with ARNN, RDE, and MT-GPRM concerning the volume of the training sample, see Fig.7 in the main text.

Fig. 7 Comparison with classic predictive models, including ARIMA, SVM, MVE,

LSTM, RC, ARNN, RDE, and MT-GPRM. We compare the robustness of the one-step ahead prediction of variable z from the 3-dimensional Lorenz system. Many methods show reliable predictions when inputting long training samples, while our FRMM remains reliable when training on short samples and verifying on long samples. η gives the proportion of the training sample.

We add comparisons with advanced delay embedding-based methods, see Tab.1 and Fig. 7 in the main text.

3. In the introduction section, the author emphasizes the necessity of predicting all networked components in a complex system and underscores "computational cost". How does FRMM achieve computational cost efficiency compared to other methods? For each x_i that needs prediction, a function ψ_i must be trained. What is the fundamental difference compared to the approach "regarding all the remaining variables as predictors"? Please elaborate further based on comment 2.

Response: Thanks for your comments and suggestions, First, we discuss the computational cost of the FRMM framework. Second, we discuss the approach "regarding all remaining variables as predictors".

(a) For the computational cost, we compare our FRMM with a representative delay embedding-based framework (e.g., ARNN). Our FRMM framework is integrated with three techniques, i.e., delay embedding, feature embedding, and Gaussian process regression. Delay embedding is used to construct a simple matrix from observed time series, whose computational cost is even low and can be negligible. For the feature embedding, our five algorithms both consist of three steps, where the first and second steps are common: the first step incorporates neighborhood information to construct a weighted graph and the third step is a spectral embedding step that involves an eigenequation computation. The second step is specific to the algorithm, transforming the neighborhood graph into suitable input for the spectral embedding step. These algorithms can be of two different kinds, either local methods (focus on the local geometry, e.g., LLE, Laplacian, Diffusion map, and LTSA) or global methods (focus on the global geometry, e.g., ISOMAP). Here, we first discuss the cost of individual representative algorithms, e.g., ISOMAP and LLE.

The ISOMAP algorithm consists of three steps: i) Finding the nearest neighbor for each data point based on the distance, the cost is approximately $O(N \log(K)L \log(L))$, where K represents the number of nearest neighbor (we set $K = 8$ for all datasets in this work), N is the input dimension and L is the length of the series. ii) Computing the

shortest path of graph based on Dijkstra's algorithm, which takes a cost $O(L^2(K + \log(L)))$. iii) Finding low-dimensional embedding via eigenvalue decomposition, it has a cost $O(EL^2)$. Thus, the cost of ISOMAP is approximately $O(N \log(K)L \log(L)) + O(L^2(K + \log(L))) + O(EL^2)$.

The LLE algorithm also has three steps: i) Searching nearest neighbors, the cost is approximately $O(N \log(K)L \log(L))$. ii) Constructing a weighted matrix, the cost is approximately $O(NLK^3)$. iii) Finding low-dimensional embedding via eigenvalue decomposition, it has a cost $O(EL^2)$. Then, the total cost of LLE is approximately $O(N \log(K)L \log(L)) + O(NLK^3) + O(EL^2)$.

The computational complexity of generic Gaussian process regression is approximately $O(N^3)$, where N represents the input dimension. In our framework, we set the E -dimensional feature manifold as a predictor, thus the complexity should be $O(E^3)$.

Thus, the total cost of FRMM:

$$C(FRMM(ISOMAP)) = O(N \log(K)L \log(L)) + O(L^2(K + \log(L))) + O(EL^2 + E^3) \quad (S9)$$

$$C(FRMM(LLE)) = O(N \log(K)L \log(L)) + O(NLK^3) + O(EL^2) + O(E^3) \quad (S10)$$

In a high-dimensional system, we often assume that $K \ll N, E \ll N$. Then, the cost can be approximated as

$$C(FRMM(ISOMAP)) = O(NL \log(L) + L^2(1 + \log(L)) + L^2) \quad (S11)$$

$$C(FRMM(LLE)) = O(NL \log(L) + NL + L^2) \quad (S12)$$

When the length of the time series is larger than the number of observed variables, i.e., $L \geq N$, we have

$$\frac{L^2(1 + \log(L))}{NL} > 1 \quad (S13)$$

Then, we infer

$$O(NL \log(L) + L^2(1 + \log(L)) + L^2) > O(NL \log(L) + NL + L^2) \quad (S14)$$

$$TC(FRMM(ISOMAP)) > TC(FRMM(LLE)) \quad (S15)$$

Thus, the combination of ISOMAP has a higher computational cost than the combination of LLE. ISOMAP is a global approach, which searches nearest neighbors between all the pairs. It would have a higher cost compared with other local approaches.

ARNN framework is an advanced framework that achieves reliable predictions in high-dimensional systems, which is an integration of delay embedding and a neural network (i.e., reservoir computing algorithm). The main computational cost of ARNN

is from the neural network. Suppose there are s neurons in the neural network, and one needs to conduct n iterations to solve the spatiotemporal transformation equation. The total cost is approximately (Ref.15 in main text)

$$C(\text{ARNN}) = O(2Nn(\frac{2}{3}L^3 + 2L^2) + s^3) \approx O(NnL^3 + L^2 + s^3) \quad (\text{S16})$$

In this work, $L \geq 200, N \geq 3$, we deduce

$$N + 2 + \log(L) < N + 2 + L < NnL + 1 + \frac{s^3}{L^2} \quad (\text{S17})$$

According to Eq. (S11), Eq. (S16) and Eq.(S17), we obtain the form (S18)

$$O(NL \log(L) + L^2(1 + \log(L)) + L^2) < O(L^2(N + 2 + \log(L))) < O(L^2(NnL + 1 + \frac{s^3}{L^2})) \quad (\text{S18})$$

Thus, FRMM has a lower computational cost than ARNN. In fact, FRMM operates without time-consuming iterations.

(b) For the difference of the approach “regarding all remaining variables as predictors”. we compare two aspects, including the computational cost and reliability. In an N -dimensional system, if we utilize Gaussian process regression to find the mapping between predictors and targets, the cost is approximately $O((N-1)^3)$ when regarding all remaining variables as predictors, and the computational cost will increase sharply, especially for a high-dimensional input. Moreover, this approaches often achieves multistep ahead prediction based on iteration, which increases the costs. Thus, using a low-dimensional feature manifold as an input takes lower computational cost when training Gaussian process regression ($O(E^3), E \ll N$). However, if we use some traditional regression-based methods (e.g., linear regression), the computational costs are also in low levels even setting all remaining variables as predictors. Thus, it's not a rigorous expression.

However, it is widely reported that real-world high-dimensional systems often contain redundant information (e.g., noise and irrelevant variables to the target variable) (Ref. 8 in main text), which may negatively affect the prediction performance. Our FRMM focuses on fundamental features of a high-dimensional system and sets low-dimensional feature manifold as predictors, which theoretically decreases some negative impacts and will be benefit for the prediction, and this is also supported by comparisons experiments, see Table 1 in mani text.

We add the discussion of computational cost in *SI Appendix Chapter 1.4*. Since

the expression ‘it has higher cost when regarding all remaining variables as predictor’ is not rigorous, we delete it and update as:

If regarding all the remaining variables as predictors, some redundant information may negatively affect the performance (e.g., noise and irrelevant variables to the target variable), especially for high-dimensional real-world systems [8].

4. The author claims that FRMM can "predict all network components," which is the most confusing point for me. I initially thought the author was going to establish a network in advance, but neither the manuscript nor SI explicitly defines what the network is.

Response: Sorry for our unclear depiction. Unlike “*network*”, the word “*networked*” is utilized as an adjective here, we have two reasons for this issue.

(a) This work aims to achieve prediction in complex dynamic systems. We use “*networked*” to depict basic characteristics of dynamic systems that system components are interacted by certain rules. Theoretically, there would be an underlying network topology that can be constructed via the interactions among components within the systems. For example, one can find a network topology of the 3-dimensional Lorenz system via component interactions, see Fig. SS1. Thus, components are intrinsically *networked* by certain rules in dynamic systems.

(b) For real-world datasets, one needs to empirically select a subsystem. In dynamical systems theory, causally linked variables share a common attractor (Ref. 28 in main text). If a variable is completely unrelated to other variables in a given subsystem, it is risky to add it into a given subsystem. Theoretically, this unrelated variable may not share a common attractor with other variables and they belong to different dynamical systems (e.g., the EEG signals from two unrelated people have no obvious interactions, they may belong to two individual dynamical systems rather than a same system), thus its time series may not be used to reconstruct an isomorphic attractor with the given subsystem by delay embedding. It is better to select a real-world subsystem whose components are *networked*.

Fig. SS1 The network topology of the 3-dimensional Lorenz system, where nodes are connected by the interactions between components.

We add these explanations in *SI Appendix Chapter 1.5*.

5. The paper combines delay embedding and feature embedding to propose FRMM. Can a dissection experiment be designed to confirm the necessity of FRMM components?

Response: Thanks for your comments, it is necessary to discuss the combination of delay embedding and feature embedding. To this end, we compare FRMM with two separate approaches, i.e., a combinative framework of delay embedding and Gaussian process regression, and a combination of feature embedding and Gaussian process regression. We first discuss the theoretical foundation of separate approaches. Then, we add new experiments to make comparisons of separate approaches.

(a) If we use feature embedding alone (including Gaussian process regression), the prediction task is operated as many generic regression approaches, which first identify optimal features as predictors by feature embedding. Then, one can conduct predictions by training correlations between input features and target variables. For example, $f(Y_t) = y_{t+T}$, Y consists of the optimal features (series) from the original system, y is a target variable, T is the predicted step. Such a framework may show limitations for multistep predictions. This framework often outputs multi-step-ahead predictions indirectly, e.g., iteration (predicted values are used as input to find longer predictions), which means that we need to conduct T experiments for T step predictions, where accumulated errors may lead to poor performance with the increase of predicted horizons. Also, multiple training processes often need relatively higher time cost. However, FRMM enables us to conduct multi-step-ahead predictions directly through the combination of delay embedding.

(b) If we use delay embedding alone (including Gaussian process regression), which has been discussed in our recently published work (“A novel framework for direct multistep prediction in complex systems” <https://doi.org/10.1007/s11071-023-08360-7>). From the combination of delay embedding and Gaussian process regression, one can construct a prediction framework that builds a mapping from the original system to the reconstructed manifolds, see Eq. S18.

$$\psi_i \begin{pmatrix} x_1(1) & x_2(1) & \cdots & x_N(1) \\ \vdots & \vdots & \cdots & \vdots \\ x_1(h) & x_2(h) & \cdots & x_N(h) \\ \vdots & \vdots & \cdots & \vdots \\ x_1(L) & x_2(L) & \cdots & x_N(L) \end{pmatrix} = \begin{pmatrix} x_i(1) & x_i(1+\tau) & \cdots & x_i(1+(E-1)\tau) \\ \vdots & \vdots & \cdots & \vdots \\ x_i(h) & x_i(h+\tau) & \cdots & x_i(L) \\ \vdots & \vdots & \cdots & \vdots \\ x_i(L) & x_i(L+\tau) & \cdots & x_i(L+(E-1)\tau) \end{pmatrix} \quad (\text{S19})$$

where the left matrix represents the points in the original system and the right matrix is identified by delay embedding from a target time series. $h = L - (E - 1)\tau$ and L is the length of the time series. This framework has been certified reliable in various systems with several components. However, it shows limitations for some high-dimensional systems, since it is not always guaranteed to conduct an isomorphic topology with an even larger embedding dimension. Moreover, high-dimensional real-world systems often show redundant information that may negatively affect the predictions. In addition to the theoretical explanations, we apply the 90-dimensional Lorenz system as an example, the combination of delay embedding and Gaussian process regression (single Gaussian process regression) exhibits poor performance, see Fig. S13.

In summary, the integration of delay embedding and feature embedding overcomes the curse of dimensionality and helps to directly achieve accurate predictions for all components in complex systems.

Fig. S13 The performance of 90-dimensional ordinary Lorenz system. (a) We use a predictive model with a combination of delay embedding and Gaussian process regression, where all the variables are used as predictors ($E = 11, \tau = 1$). (b) The prediction results via a combinative model from feature embedding and Gaussian

process regression. We first select 11 ($E=11$) features of the Lorenz system via diffusion map, and then we operate predictions by training correlations between these features and a target variable. Results show that neither feature embedding nor delay embedding framework can be separately used for reliable predictions of all components in complex systems, while the integration of delay embedding and feature embedding (FRMM) has the potential to address that prediction task (see Figs. 3 (d-f) in main text).

We add these discussions in *SI Appendix Chapter 1.6*.

6. It is suggested to integrate Fig. S1 with Fig. 2 and provide a clearer FRMM framework in the main text. Besides, what do the different colors of points in Fig. 1 represent?

Response: Thanks for your suggestions.

i) We have integrated Fig. S1 and Fig. 2 and reorganized a clearer FRMM framework, see Fig. 2 in the main text. Then, we delete Fig. S1 in *Appendix*.

ii) There are no typical implications of different colors in Fig. 1. To avoid misleading, we have constructed a more natural one, see Fig. 1 in main text.

Fig. 2 Sketch of FRMM framework. To forecast all components in an N -dimensional system (a), we find its E -dimensional ($E \ll N$) representations from delay embedding (b) and feature embedding (c). Thus, an N -dimensional dynamical system M is represented by two isomorphic low-dimensional manifolds (i.e., feature manifold M_0 and reconstructed manifold M_{x_i}). The foundation of an isomorphism enables the feature manifold M_0 to be utilized as a universal predictor to find all the unknown elements (shadow elements in the matrix M_{x_i} represent the future dynamics of time series x_i) on every reconstructed manifold M_{x_i} (d). Once identify the mapping ψ_i ($i = 1, 2, \dots, N$), we can find the future dynamics of all components in complex systems.

Fig. 1 Low-dimensional embeddings of complex systems. The dynamical structure of the 3-dimensional Lorenz system (a) is represented in 2-dimensional space via delay embedding (b) (from time series x , where $E = 2, \tau = 10$) and feature embedding (c) (i.e., diffusion map algorithm). Analogously, one can find 2-dimensional representations (e and f) of the 3-dimensional Rossler system (d). Even with additive noise, one can also find their low-dimensional embeddings (cf. *SI Appendix* Fig. S1).

We update Fig.1 and Fig. 2 in main text and delete Fig. S1.

7. The Takens' embedding theory is usually used in a steady state, but the EGG data is non-stationary, how does the author predict non-stationary EGG data? Is preprocessing needed? Is denoising necessary?

Response: Thanks for your comments. As you mentioned that delay embedding is usually used in steady states, while the EGG signals show non-stationary characteristics. Our experiments show that FRMM can address predictions for raw EEG series, it may depend on the selected samples. To test FRMM, we select other signal series from a heart-lungs-blood oxygen concentration system (a physiological system). Three signal series (i.e., heart rate, chest volume, and blood oxygen concentration) are free from <https://www.physionet.org/content/santa-fe/>. Similarly, these series are also non-stationary, but FRMM fails to achieve their accurate predictions from raw series. Following your suggestion, we normalize the raw series by Z-Score normalization. The results show that FRMM can achieve accurate ahead predictions for all series in the heart-lungs-blood oxygen concentration system (see Fig. SS2). Still, the average performance is improved obviously when normalizing the EEG series (the metrics from

raw series are $\rho = 0.89$ and $RMSE = 0.6$, while performances from normalized series are $\rho = 0.92$ and $RMSE = 0.42$). We update all results for the EEG series, see Figs. 4(d-f) in main text

Fig. SS2 The performance of the heart-lungs-blood oxygen concentration system. The series are preprocessed by Z-Score normalization.

Fig. 4 Real-world samples, including the daily wind speed (m/s) (a-c), per second EEG signal (d-f), daily exchange rate (g-i), and traffic speed (5-minute interval) (j-l). By randomly selecting 50% of the data as a test sample, the FRMM is shown reliable for accurate multistep predictions in representative real-world systems, where the predicted horizons are $T = 20$ (a-i) and $T = 10$ (j-l).

We update the predictions of the EEG series in the main text, see Fig. 4 in the main text and Fig. S5 in *Appendix*. And we note the normalization process for EEG datasets, see *SI Appendix Chapter 1.2.3*.

8. Concerning different data, is there a criterion for choosing which dimension reduction method for feature embedding to use? How did the author determine which method to adopt?

Response: Thanks for your comments. In this work, we use several classic manifold learning algorithms to find faithful low-dimensional representation of a high-dimensional system. A fundamental basis of these algorithms is to preserve geometry features of the original system in a low-dimensional space, e.g., the nearest neighbors in the original system are also nearest neighbors in its low-dimensional representation. However, real-world systems may show different topological structures and then exhibit quite different geometry features, there is no well-accepted algorithm that is faithful for all the systems. On this basis, several advanced feature embedding algorithms are proposed for different geometries or similar geometries but with different aspects, e.g., ISOMAP, LLE, Laplacian, Diffusion map, and LTSA. ISOMAP finds low-dimensional representation based on global geodesic distance (different from Euclidean metric) and is available for the manifold as a convex region distorted in certain ways, such as folding or twisting. Unlike ISOMAP, LLE, Laplacian, Diffusion map, and LTSA focus on local geometry. LLE assumes that data on a manifold can be approximated by linear combinations of their nearest neighbors, it performs well when the data are uniformly sampled over the manifold but shows limitations for some closed manifolds, e.g., sphere and torus. Laplacian algorithm is closely related to LLE, which measures the local feature by an isotropic diffusion kernel. Diffusion map approximates local geometry by diffusion distance. LTSA utilizes the tangent space in the neighborhood of a data point to represent the local geometry.

Still, it is challenging to give reliable criteria for the selection of algorithm without enough prior knowledge of system topological structure (see Ref.1 in *Appendix*), especially in real-world systems, only from observed time series, it is not easy to view their geometries in a high-dimensional space. Nevertheless, the selected techniques make sense for many systems. In practical applications, it is possible to compare them and find an optimal one, since these algorithms are computationally efficient that they often use eigenvalue decomposition without complex iterative processes. Take the 90-dimensional Lorenz system as an example, we calculate the time costs for different

feature embedding algorithms. All the algorithms show low computational costs, see Table S3.

Table S3. Time costs of different feature embedding algorithms

Algorithm	Time cost (Second)
ISOMAP	12.3919
LLE	0.32629
Laplacian	0.14581
Diffusion map	0.29983
LTSA	0.21549

Note: We identify the low-dimensional ($E = 11$) representation of the 90-dimensional Lorenz system via different algorithms. All algorithms show relatively low time costs.

We add these discussions in *SI Appendix Chapter 1.7*.

Additional modification: Based on the regulation of NC, we shorten the Abstract with no more than 150 words, as follows:

Forecasting all networked components in complex systems is an open and challenging task, possibly due to high dimensionality and undesirable predictors. We bridge this gap by proposing a data-driven and model-free framework, namely, feature-and-reconstructed manifold mapping (FRMM), which is an innovative combination of feature embedding and delay embedding. For a high-dimensional dynamical system, FRMM finds its topologically equivalent manifolds with low dimensions from feature embedding and delay embedding and then sets the low-dimensional feature manifold as a universal predictor to achieve predictions of all networked components. The substantial potential of FRMM is shown for both representative models and real-world data involving Indian monsoon, electroencephalogram (EEG) signals, foreign exchange market, and traffic speed in Los Angeles Country. FRMM overcomes the curse of dimensionality and finds a generalized predictor, and thus has great potential for applications in many other real-world systems.

Reviewer #2 (Remarks to the Author):

Authors proposed a data-driven and model-free framework, FRMM, by combining feature and delay embedding to predict all components of a complex networks. It is claimed that this is a universal predictor and can be applied to high to low dimensional

dynamical systems and verified by real data. In the last two decades, it was an open problem to design a generalized framework for the predictions of all networked components in a complex system.

The results are interesting and demand for publication after a minor editing:

Response: Thanks for your positive evaluation of our work. Your constructive comments and suggestions substantially help to improve our work.

Note that Fig. SS1, Fig. SS2, and Fig. SS3 are used to explain related issues in this letter and not be contained in our article. The modified contents are noted by blue color, see attached *Revised manuscript with notes* and *Revised SI with notes*. Also, we give clear main manuscript and SI.

1. In delay embedding, is there any methods to choose delay τ ? In result section, $\tau=10$ was chosen. Also the choice of the parameter E .

Response: Thanks for your comments. Both embedding dimension (E) and time delay (τ) are important parameters to conduct delay embedding. Theoretically, an isomorphic reconstruction is guaranteed on the condition $E > 2d$, where d represents the box dimension of the attractor. The estimated box dimensions in selected systems are provided in *SI Appendix* Table S4. In this work, we use the false nearest neighbor (FNN) and mutual information to determine the embedding dimension and time lag, respectively. In practice, the selection of time lag is flexible that several different lags can be used to find available reconstruction, see *SI Appendix* Fig. S9. Note that the predicted step is determined by $T = (E - 1)\tau$, we test the predicted horizons by increasing the time lags. However, it is still challenging to achieve reliable even longer predictions in complex systems, since the reconstructed attractor may exhibit a completely different topology from the original system when applying a large time lag.

Table S4. The estimated box dimension in selected systems

Objection	Embedding dimension (E)	Box dimension (d)	Feature embedding
The 3-dimensional Lorenz system	2	0.91	Diffusion map
The 3-dimensional Rossler system	2	0.92	LTSA
The 90-dimensional ordinary Lorenz system	11	1.51	Diffusion map

The 90-dimensional Lorenz system with varying dynamics	11	1.59	Diffusion map
The 231-dimensional wind speed system	5	1.68	LLE
The 64-dimensional EEG signal system	5	1.74	LTSA
The 70-dimensional exchange rate system	5	1.36	ISOMAP
The 46-dimensional stock price system	5	1.33	Laplacian
The 207-traffic speed system	10	1.98	Laplacian

Note: The box dimension is estimated by the FRACLAB Toolbox in MATLAB R2018a.

We add these contents in *SI Appendix Chapter 1.8*.

2. In Fig. 1, the axes labels are missing.

Response: Sorry for our mistakes, we have modified it. To avoid misleading by the complex colors, we give a more natural one, see Fig.1 in main text.

Fig. 1 Low-dimensional embeddings of complex systems. The dynamical structure of the 3-dimensional Lorenz system (a) is represented in 2-dimensional space via delay embedding (b) (from time series x , where $E = 2, \tau = 10$) and feature embedding (c) (i.e., diffusion map algorithm). Analogously, one can find 2-dimensional representations (e and f) of the 3-dimensional Rossler system (d).

3. In Eq. (14), why the parameter $b=-8/3$? In Lorenz system the parameter b is positive, I think this is a typo mistake. Please check.

Response: Sorry for this mistake, we have checked and modified it, see Eq.14 in *Materials and methods*.

$$\begin{aligned}\dot{x}_i &= \sigma(t)(y_i - x_i) + cx_{i-1}, \\ \dot{y}_i &= ax_i - y_i - x_iz_i, \\ \dot{z}_i &= -bz_i + x_iz_i,\end{aligned}\tag{14}$$

where $\sigma(t)$ is the time-varying parameter. a and b are set to be generic values, i.e., $a = 28, b = 8/3$.

4. In real systems, one cannot avoid the presence of noise. Is the proposed method applied for the presence of small noise? Discussion on this is needed.

Response: Thanks for your comments. We add new experiments to discuss the robustness of noise. First, we test the efficiency of low-dimensional embeddings from noise systems. Results show that it is still possible to find low-dimensional representations from delay embedding and feature embedding even with additive noise, see Fig. S2. Second, we also conduct experiments to discuss the performance with deteriorating noise (e.g., $\sigma = 0.1, 0.2, \dots, 0.9$), see Fig. 5(b) in main text (Discussion). The results show robustness of our FRMM framework.

Fig. S1 Low-dimensional embedding of noisy systems. Based on delay embedding and feature embedding, one can find low-dimensional representations of the Lorenz system and Rossler system even with additive noise ($\sigma = 0.5$, σ represents noise strength).

The discussions of noise are given in Fig. 5(b) in main text. We add the simulation of low-dimensional embeddings from noisy systems, see Fig. S1 in *Appendix*.

5. The figure caption should be in details, at least the finding of main results.

Response: Thanks for your suggestions, we have checked all the figures and added more details, especially the explanations and mains results.

6. Authors verified the proposed techniques for two model simulations and 4 real data. It is claimed that the predictor is universal. Why is it universal? Will it be applied for other systems and other network topologies?

Response: As you mentioned, the universality of the identified predictor should be furtherly discussed. To this end, we first add four new systems to support our FRMM, including two model systems (i.e., 3-dimensional Chen chaotic system (Eq. S19) and 3-dimensional ecology system (Eq. S20)) and two real-world systems (i.e., the series from a physiological system and the index and sea surface temperature in ENSO cycle). Then, we provide mathematical explanations to explain the universality of selected predictor.

(a) The performance of new model systems

Chen system is another classic chaotic system, see Eq. (S19). With an initial state $(x(0), y(0), z(0)) = (1, 2, 3)$ and integrated interval $[0, 29]$, we obtain three time series. Each of them has a length of 1557. FRMM yields accurate predictions for all the components (Fig. S14(a)).

$$\begin{cases} \dot{x} = c_1(y - x), \\ \dot{y} = c_2x(1 + z) + c_3y, \\ \dot{z} = xy - c_4z, \end{cases} \quad (\text{S19})$$

We have tested the performances in several chaotic systems, here we discuss a generic nonlinear system (non-chaotic system), i.e., the 3-dimensional ecology systems (see Eq. (1)). From the initial state $(x(0), y(0), z(0)) = (0.4, 0.5, 0.6)$ and time interval $t = 1, \dots, 399$, we obtain times series with length 400. By setting the feature manifold as a predictor, we achieve accurate ahead predictions for all variables, see Fig. S14 (b).

$$\begin{aligned} x(t+1) &= x(t)[4 - 4x(t) - 2y(t) - 0.4z(t)], \\ y(t+1) &= y(t)[3.1 - 0.31x(t) - 3.1y(t) - 0.93z(t)], \\ z(t+1) &= z(t)[2.12 + 0.636x(t) + 0.636y(t) - 2.12z(t)], \end{aligned} \quad (\text{S20})$$

Fig. S14 The performance in the 3-dimensional Chen system (a) and an ecology system (b). By setting the feature manifold as a predictor, we achieve accurate prediction for all components in complex systems. Note: we validate FRMM by randomly selecting 50% of the series as the test sample. Diffusion map is utilized to find the feature manifold, and $E = 2, \tau = 8$.

(b) The performance in real-world systems

Fig. S15 The performance in heart-lungs-blood oxygen concentration system (a-c) and ENSO cycle (d-k). By setting their feature manifolds as generalized predictors, we make predictions for all components in these systems. Note: Heart-lungs-blood oxygen concentration system contains three time series variables, i.e., heart rate, chest volume, and blood oxygen concentration. Due to different units, we normalize all the series against their mean and standard deviation. The ENSO cycle consists of three anomaly series (index) and three sea surface temperature (SST) series in ENSO 1+2, ENSO 3, ENSO 3.4, and ENSO 4 regions. k represents the randomly selected test sample (50% of the entire series).

(c) Mathematic explanations

Given an N -dimensional dynamical system with original manifold M , FRMM aims to find its low-dimensional representations from different approaches.

According to the manifold learning technique (feature embedding), for each state

point on M , we can find a corresponding state point in a low-dimensional representation M_0 ($M_0 \subseteq R^E, E \prec N$) via a mapping ϕ , and this mapping exists and homeomorphism (see Ref. 31 in the main text). This process can be given mathematically, $\forall X(t) \in M$, $Y(t) \in M_0$, we have (S21)

$$Y(t) = \phi(X(t)), t = 1, 2, \dots, L. \quad (\text{S21})$$

Since ϕ is a homeomorphism, we deduce an inverse form (S22)

$$X(t) = \phi^{-1}(Y(t)). \quad (\text{S22})$$

On the other hand, in a dynamical system, each time series variable can be utilized to reconstruct an isomorphic manifold M_{x_i} ($M_0 \subseteq R^E, E \prec N$) based on delay embedding theory. This process can be also given mathematically, $\forall X(t) \in M$, $\tilde{X}_i(t) \in M_{x_i}$, we have (S23)

$$\tilde{X}_i(t) = \varphi_i(X(t)), t = 1, 2, \dots, L, i = 1, 2, \dots, N. \quad (\text{S23})$$

Considering Eq. (S22) and Eq. (S23), we obtain (S24)

$$\tilde{X}_i(t) = \varphi_i(\phi^{-1}(Y(t))). \quad (\text{S24})$$

Thus, $\forall \tilde{X}_i(t) \in M_{x_i}$, $Y(t) \in M_0$, we have (S25)

$$\psi_i(Y(t)) = \tilde{X}_i(t), i = 1, 2, \dots, N, \quad (\text{S25})$$

where $\psi_i(x) = \varphi_i \phi^{-1}(x)$.

Consequently, we obtain (S26)

$$\psi_1(Y(t)) = \tilde{X}_1(t), \dots, \psi_N(Y(t)) = \tilde{X}_N(t), \quad (\text{S26})$$

where $Y(t) \in M_0$ and $\tilde{X}_i(t) \in M_{x_i}$.

From Eq. (S26), we conclude that the feature manifold M_0 is possible to select as a generalized predictor and is available to predict future dynamics for all components in complex systems once finding the mapping ψ_i .

We add these discussions in SI Appendix Chapter 1.9.

Overall, I have enjoyed the reading of the results and the results are really interesting. Before publication, the above points may be discussed.

Response: Thanks again for your comments and suggestions, we have improved the work as much as possible. Hope that we have addressed all your concerns.

Additional modification: Based on the regulation of NC, we shorten the Abstract with no more than 150 words, as follows:

Forecasting all networked components in complex systems is an open and challenging task, possibly due to high dimensionality and undesirable predictors. We bridge this gap by proposing a data-driven and model-free framework, namely, feature-and-reconstructed manifold mapping (FRMM), which is an innovative combination of

feature embedding and delay embedding. For a high-dimensional dynamical system, FRMM finds its topologically equivalent manifolds with low dimensions from feature embedding and delay embedding and then sets the low-dimensional feature manifold as a universal predictor to achieve predictions of all networked components. The substantial potential of FRMM is shown for both representative models and real-world data involving Indian monsoon, electroencephalogram (EEG) signals, foreign exchange market, and traffic speed in Los Angeles Country. FRMM overcomes the curse of dimensionality and finds a generalized predictor, and thus has great potential for applications in many other real-world systems.

REVIEWER COMMENTS

Reviewer #1 (Remarks to the Author):

It is good that the authors addressed the points that concerned me, point by point. I appreciate the authors for incorporating and implementing some of my comments from my last review. However, from my point of view, there are still issues that have not been addressed. I hope to see the revised manuscript with the following concerns fully resolved and greatly improved to meet the high standards of Nature Communications.

1. The authors stated “Two accuracy metrics are employed, including (Pearson correlation between predicted values and observed values) and RMSE (Normalized by the standard deviation of the input series)”. However, what do the numbers in Table 1 signify? Why does the performance decrease as the training sample size decreases and the test sample size increases? It is not clear why 'FRMM is robust' from Table 1. Can the authors provide some explanations?"
2. As for the statement, “predicting all networked components in a complex system”, the authors should clarify whether FRMM can predict all networked components “synchronously” or not. To my understanding, FRMM still uses a framework 'for each x_i that needs prediction, a function ψ_i must be trained.' Thus, this is fundamentally the same as other prediction methods based on STI. If that's the case, FRMM is merely another framework that may have some improvements in terms of time cost. The authors should discuss more about this fundamental issue.
3. Anyway, I was pleased that the authors compared the time complexities of FRMM with other prediction methods. In ARNN, 'L' represents the embedding dimension (with the prediction length being L-1), rather than the length of the series, and $L \ll n$. Therefore, the assumption 'When the length of the time series is larger than the number of observed variables' may not hold. In this situation, there could be an issue with Eq. (S18). The authors should provide a more accurate analysis of this matter.
4. Although variables from some representative models like a Lorenz system are networked and correlated, the network structure for many real-world datasets is unknown. Therefore, I don't believe that the statement 'predict all networked components' in the title is appropriate, as the authors do not explain any causal relations or interactions among variables in the manuscript.
5. In the Method section, there is a crucial inconsistency in the FRMM framework. From Eq. (11), $\psi_i: R^{(L \times E)} \rightarrow R^{(L \times (1 + (E-1)\tau))}$, however, in Eq. (12), $\psi_i: R^{(L \times E)} \rightarrow R^{((L + (E-1)\tau) \times 1)}$. Please check which one is correct. Also, as per the guidelines of Nature Communications, the authors should publicly share the source code of FRMM and the data used on a platform such as GitHub.
6. The author updated the colors of the points with blue curves. However, different colors of points in Fig. 6 are still being used without any explanation. I think the authors should pay more attention to these details and double-check them.

Reviewer #1 (Remarks on code availability):

The authors didn't provide the codes of the proposed method.

Reviewer #2 (Remarks to the Author):

The authors have revised the paper nicely, I am happy to recommend it for publication in Nat Comm.

Reviewer #1 (Remarks to the Author):

It is good that the authors addressed the points that concerned me, point by point. I appreciate the authors for incorporating and implementing some of my comments from my last review. However, from my point of view, there are still issues that have not been addressed. I hope to see the revised manuscript with the following concerns fully resolved and greatly improved to meet the high standards of Nature Communications.

Response: Thank you very much for your positive reply to our efforts. Still, thanks for your constructive comments and suggestions, which help to make our work even stronger. More importantly, many of your suggestions inspire most of us for our future research.

We have checked the whole work carefully and modified all the shortages as much as possible, hope to address all your concerns and meet the standards of *Nature Communications*.

Note that Fig. SS1 is used to explain related contents, which is not contained in our article. The modified contents are noted in blue color, see the attached *Revised manuscript with notes* and *Revised SI with notes*. Also, we give a clear main manuscript and SI.

1. The authors stated “Two accuracy metrics are employed, including (Pearson correlation between predicted values and observed values) and RMSE (Normalized by the standard deviation of the input series)”. However, what do the numbers in Table 1 signify? Why does the performance decrease as the training sample size decreases and the test sample size increases? It is not clear why 'FRMM is robust' from Table 1. Can the authors provide some explanations?"

Response: Thanks for your comments. As you mentioned, there are some unclear and inaccurate descriptions for Table 1, as well as its related contents.

1) In Table 1, for each dataset, the number in the first row represents the metric ρ (the Pearson correlation between predicted values and observed values), and the number in the second row gives the error (RMSE). We add clear explanations of Table 1.

2) For “the performance decreases as the training sample size decreases and the test sample size increases”, we first give an explanation, and then we conduct experiments to support this issue. Generally, longer series often contain more information and are beneficial for training more suitable mappings or correlations between series. Then, it

may yield better performance when inputting longer training samples. This is common for many artificial neural networks. Conversely, we may find incorrect or poor mappings between series when inputting short series, then it is possible to output more errors when testing on longer series. Take the 90-dimensional ordinary Lorenz system as an example, the results show that the performance of FRMM decreases as the training sample length decreases and the test sample lengths increase, but FRMM remains robust even with a short training sample, see Fig. S11 in *Appendix*. Maybe the “sample size” has misleading meanings since it can be used to describe the number of variables. So, we replaced it with the “sample length” in our article.

3) For the 'FRMM is robust' from Table 1, we are sorry for this inaccurate interpretation. Table 1 gives the average performance of several real-world datasets compared with classic predictive methods. It shows that FRMM has relatively better performance across datasets in total, which fails to discuss any information for robustness. In our work, we first compare the **robustness** with several classic methods concerning the length of training and test sample (Fig. 7). Then, we compare the **performance** across different real-world datasets (Tab. 1). we clarified and modified these contents in the main text.

We first add clear descriptions for the number in Table 1. Then, the robustness test for the length of the training sample and test sample is given in the *Robustness tests* in the *Discussion* in the main text. Finally, we clarify the logic of the contents related to Fig. 7 and Tab. 1.

Fig. S11 The robustness test on the length of the training samples. Our framework is robust with short and longer training series. η ($0.1 \leq \eta \leq 0.9$) represents the proportion of the training sample. The 90-dimensional ordinary Lorenz system is used as an example.

2. As for the statement, “predicting all networked components in a complex system”, the authors should clarify whether FRMM can predict all networked components “synchronously” or not. To my understanding, FRMM still uses a framework 'for each x_i that needs prediction, a function ψ_i must be trained.' Thus, this is fundamentally the same as other prediction methods based on STI. If that's the case, FRMM is merely another framework that may have some improvements in terms of time cost. The authors should discuss more about this fundamental issue.

Response: Thanks for your comments. Indeed, the STI equation gives a fantastic framework to transform spatial states to a temporal series. Several advanced STI-based predictive methods have been proposed like RDE, MT-GPRM, and ARNN. FRMM is another STI-based framework, which has other differences and improvements

comparing with many existing STI-based methods, in addition to the time cost (the discussion of time cost is provided in *SI Appendix* Chapter 1.4.). We clarify and highlight the main improvements from three aspects, including the prediction task, the architecture, and the theoretical foundation.

For the prediction task, existing STI-based methods show high reliability for several significant tasks, e.g., one-step and multistep ahead predictions. However, the prediction task of all components in a complex system is an important issue (we have explained the significance in *Introduction* of the main text, see lines 54-68), but this issue is still unsolved entirely. Though the existing STI-based frameworks have the potential to address this issue, their certifications are often based on partial components, their abilities are not fully certified and tested for all components. This may bring misleading and even false conclusions. We give a simple example to explain it. For the generic 3-dimensional Lorenz system that consists of x , y , and z . It is possible to use a traditional linear regression model to achieve accurate predictions for x and y . However, it fails to predict variable z , see Figure S11. In fact, x and y show a highly positive correlation from their time series, but there is no obvious linear correlation between x and z . It is uncritical to conclude that this linear regression model can make accurate predictions for the Lorenz system. Thus, examining the reliability of a predictive model for complex systems only by several tests on partial components may be risky. We have emphasized this fundamental issue as the third challenge in *Introduction* of the main text, “(c) *The challenge in forecasting multiple observations typically results in verifying methods over only a single or possibly few observations*”. Thus, the prediction task for all components in complex systems is still unsolved entirely, which is refined by our FRMM framework.

For the architecture, the main difference between FRMM and other STI-based frameworks is the selection of predictor. Some existing STI-based frameworks set the original system as the predictor, e.g., MT-GPRM, some frameworks may use different predictors for different targets, e.g., for each target variable, ARNN finds several highly related components as predictor via mutual information method. FRMM focuses on system’s fundamental dynamics and sets the system’s low-dimensional feature manifold as a fixed predictor to predict all components in complex systems, which theoretically helps to reduce some negative impacts of redundant information in a high-dimensional system.

For the theoretical foundation, many existing STI-based frameworks create the STI equation by non-delay embedding and delay embedding, which originates from that a complex system can be approximately represented by different coordinates. Generally, the non-delay embedding of complex systems can be approximated in a space with either low or high dimension. (e.g., MT-GPRM sets all selected observations as a representation of original systems, while RDE finds non-delay embedding by randomly selecting several observations, and ARNN uses mutual information method to select several highly correlated variables of targets as non-delay embedding). The theoretical foundation of FRMM is based on a well-accepted report that a high-dimensional system often has redundant information, and the system's fundamental dynamics (e.g., the topology of complex systems) are restored in low-dimensional manifolds (<https://www.nature.com/articles/s41567-023-02303-0>, also see Ref. [31-35] in the main text). FRMM framework focuses on low-dimensional dynamics of complex systems, and these low-dimensional dynamics are identified by feature embedding and delay embedding. The feature embedding is conducted by powerful manifold learning algorithms, and these methods can automatically restore the fundamental topology of the original system in a low-dimensional space. Thus, FRMM has differences with existing STI-based frameworks concerning the theoretical foundations. Additionally, identifying the fundamental dynamics of a high-dimensional system theoretically helps to reduce the negative impacts of redundant information in a high-dimensional system, and will be beneficial for better predictions. These are also supported by the relatively higher performance and robustness of FRMM in many real-world datasets (Tab. 1 and Fig. 7 in main text).

In summary, in addition to the time cost and performance, FRMM has meaningful improvements compared with many existing STI-based frameworks.

However, like other STI-based methods, FRMM predicts all components one by one, that is, for each target variable, a function needs to be trained. Your comments inspire us to improve the framework to predict all components synchronously, it is an interesting and important topic, which we will try to consider in our future research.

Figure SS1. The predictions of variables x and z in the 3-dimensional Lorenz system. We can achieve faithful prediction for variable x based on a linear regression model. However, this model does not work for variable z . Thus, examining the reliability of a predictive model for complex systems only by several tests on partial components may be risky.

We add related contents to *Discussion* in the main text (lines 387-434), as follows

FRMM is developed based on a popular framework, namely spatiotemporal information (STI) transformation. Several advanced STI-based methods (e.g., RDE [14], MT-GPRM [13], and ARNN [15]) have been proposed to predict various complex systems. FRMM shows individual characteristics and meaningful improvements comparing with many existing STI-based methods. We clarify them from three aspects, including the prediction task, the architecture, and the theoretical foundation.

For the prediction task, it is still unsolved for the predictions of all components in complex systems. Though some existing STI-based frameworks have the potential to address this issue, their abilities are often certified on partial components and not fully tested by all units in complex systems. Note that verifying a predictive model on fewer observations of complex systems may be risky. Take the generic 3-dimensional Lorenz system as an example, it is possible to predict variables x and y through a linear regression model, but this model fails to predict variable z [44]. It is uncritical to conclude that a linear regression model can predict the Lorenz system. In this direction, FRMM is faithful and exhibits higher potential for the predictions of all components in complex systems.

For the architecture, the main difference between FRMM and other STI-based frameworks is the selection of predictor. Some STI-based frameworks set the original system as the fixed predictor, e.g., MT-GPRM, while some frameworks may use different predictors for different targets, e.g., for each target variable, ARNN finds several highly

related components as predictors. FRMM focuses on system's fundamental dynamics and sets the system's low-dimensional feature manifold as a fixed and generalized predictor, which gives an efficient predictor when predicting different components in complex systems.

For the theoretical foundation, many existing STI-based frameworks create the STI equation by non-delay embedding and delay embedding, which originates from that a complex system can be approximately represented by different coordinates. Generally, the non-delay embedding of complex systems can be approximated in a space with either low or high dimension. (e.g., MT-GPRM sets all selected observations as a representation of original systems, RDE finds non-delay embedding by randomly selecting several observations). The theoretical foundation of FRMM is based on a well-accepted report that a high-dimensional system often has redundant information, and the system's fundamental dynamics (e.g., the topology of complex systems) are restored in low-dimensional manifolds [31-35]. FRMM framework focuses on low-dimensional dynamics of complex systems, and these low-dimensional dynamics are identified by feature embedding and delay embedding. The feature embedding is conducted by powerful manifold learning algorithms, and these methods can automatically extract and restore the fundamental topology of the original system in a low-dimensional space. Thus, FRMM shows different theoretical foundations with existing STI-based frameworks. Additionally, identifying the fundamental dynamics of a high-dimensional system theoretically helps to reduce the negative impacts of redundant information in a high-dimensional system, and will be beneficial for better predictions. These are also supported by the relatively higher performance and robustness of FRMM in many real-world datasets (Tab. 1 and Fig. 7 in main text).

However, like other STI-based frameworks, FRMM fails to predict different components synchronously. In other words, for each target variable, one needs to train a suitable mapping.

3. Anyway, I was pleased that the authors compared the time complexities of FRMM with other prediction methods. In ARNN, 'L' represents the embedding dimension (with the prediction length being L-1), rather than the length of the series, and $L \ll n$. Therefore, the assumption 'When the length of the time series is larger than the number of observed variables' may not hold. In this situation, there could be an issue with Eq. (S18). The authors should provide a more accurate analysis of this matter.

Response: Sorry for our mistakes. We recognize the comparisons of time cost. Reservoir computing (RC) is the main basis of ARNN, and it is also a recently popular and powerful recurrent neural network for the predictions of chaotic dynamics. Thus, we compare the time cost between traditional reservoir computing algorithms and our FRMM framework. A general reservoir computing scheme consists of an input layer, a reservoir network, and an output layer (Ref. 1 in *Appendix*). When predicting T -step forward, RC first maps an N -dimensional input data to a D -dimensional vector by the $D \times N$ matrix. Then, RC conducts predictions from the reservoir to the output module by the $T \times D$ matrix. There may need M neurons in the reservoir network, and the time cost of the processing of reservoir converting is $O(D^3)$. Suppose there are S iterations from reservoir to output, the time cost for the transformation is approximately $O(2SN(2/3T^3 + 2T^2))$. The total cost of generic RC is approximately $O(2SN(2/3T^3 + 2T^2) + D^3)$.

In general, it is difficult to conduct accurate even longer predictions in chaotic systems. Thus, T is a relatively small value. For a generic RC, the reservoir has a much higher dimension (cf. 1 in *Appendix*), i.e., $N \ll D$. Thus, the main computational cost for a generic RC can be determined as $O(D^3)$.

When $L < N \ll D$, we obtain

$$O(NL \log(L) + L^2(1 + \log(L)) + L^2) < O(L^2(N + 2 + \log(L))) < O(N^2(N + 2 + N)) \quad (\text{S16})$$

$$O(N^2(N + 2 + N)) \approx O(N^3 + N^2) < O(D^3) \quad (\text{S17})$$

Generally, FRMM has a lower computational cost than the generic reservoir computing network.

We modify and reorganize some contents (above discussions) in *SI Appendix Chapter 1.4*.

4. Although variables from some representative models like a Lorenz system are networked and correlated, the network structure for many real-world datasets is unknown. Therefore, I don't believe that the statement 'predict all networked components' in the title is appropriate, as the authors do not explain any causal relations or interactions among variables in the manuscript.

Response: We agree with your comments completely. As you mentioned, we can easily find the connections of components in some model systems from their differential equations. However, it is unknown of the causation in many real-world systems, and causality inference is now a hot topic and still an open issue across many disciplines.

Though we cannot find accurate causal interactions in real-world systems, to better meet the dynamical systems theory, it is better to select a real-world subsystem, whose components are possibly and intuitively networked. Thus, we also maintain a short guidance for the selection of subsystems from real-world datasets, see *SI Appendix Chapter 1.5*.

Still, we updated a more suitable and understandable title: *Predicting Multiple Observations in Complex Systems through Low-Dimensional Embeddings*.

Moreover, to avoid misleading, we delete the “networked” in our article, except for the discussion of selection for real-world datasets in *SI Appendix Chapter 1.5*.

5. In the Method section, there is a crucial inconsistency in the FRMM framework. From Eq. (11), $\psi_i: \mathbb{R}^{(L \times E)} \rightarrow \mathbb{R}^{(L \times (1 + (E-1)\tau))}$, however, in Eq. (12), $\psi_i: \mathbb{R}^{(L \times E)} \rightarrow \mathbb{R}^{((L + (E-1)\tau) \times 1)}$. Please check which one is correct. Also, as per the guidelines of Nature Communications, the authors should publicly share the source code of FRMM and the data used on a platform such as GitHub.

Response: Thanks for your careful comments. The mappings in Eq. (11) and Eq. (12) are different. We have distinguished them with ψ_i and $\hat{\psi}_i$. The related explanations are given as follows.

In Eq. (11), ψ_i represents a mapping from the feature manifold to the reconstructed manifold. As we know, the reconstructed manifold is obtained by the delay coordinates of a given time series, i.e., $(x_i(t), x_i(t + \tau), \dots, x_i(t + (E-1)\tau))$ (the right matrix in Eq. (11)). To conduct longer horizon predictions (e.g., we can achieve at most $(E-1)\tau$ step predictions per experiment when finding $x(L + (E-1)\tau)$), we, therefore, consider the final coordinate of the reconstructed manifold, i.e., $x(t + (E-1)\tau), t = 1, 2, \dots, L$, see the blue elements of the right matrix in Eq. (11). Thus, in practical applications, we can conduct prediction by training mappings from feature manifold to the final coordinate of each reconstructed manifold. This approach is available since a one-to-one mapping between the feature manifold and reconstructed manifold is held, and then it is possible to conduct a mapping from the feature manifold to each coordinate of the reconstructed manifold. Especially, we consider the final coordinate to obtain longer predictions. We also explain these processes mathematically, see Eqs. (11-14).

$$\psi_i \begin{pmatrix} y_1(1) & y_2(1) & \cdots & y_E(1) \\ \vdots & \vdots & \cdots & \vdots \\ y_1(h) & y_2(h) & \cdots & y_E(h) \\ \vdots & \vdots & \cdots & \vdots \\ y_1(L) & y_2(L) & \cdots & y_E(L) \end{pmatrix} = \begin{pmatrix} x_i(1) & x_i(1+\tau) & \cdots & x_i(1+(E-1)\tau) \\ \vdots & \vdots & \cdots & \vdots \\ x_i(h) & x_i(h+\tau) & \cdots & x_i(L) \\ \vdots & \vdots & \cdots & \vdots \\ x_i(L) & x_i(L+\tau) & \cdots & x_i(L+(E-1)\tau) \end{pmatrix}. \quad (11)$$

According to the reconstructed manifold, we conduct a mapping from M_{x_i} to its final coordinate

$$\widehat{\phi}_i \begin{pmatrix} x_i(1) & x_i(1+\tau) & \cdots & x_i(1+(E-1)\tau) \\ \vdots & \vdots & \cdots & \vdots \\ x_i(h) & x_i(h+\tau) & \cdots & x_i(L) \\ \vdots & \vdots & \cdots & \vdots \\ x_i(L) & x_i(L+\tau) & \cdots & x_i(L+(E-1)\tau) \end{pmatrix} = \begin{pmatrix} x_i(1+(E-1)\tau) \\ \vdots \\ x_i(L) \\ \vdots \\ x_i(L+(E-1)\tau) \end{pmatrix}. \quad (12)$$

In Eq. (12), $\widehat{\phi}_i$ can be easily obtained by the transform (13)

$$\begin{pmatrix} x_i(1) & x_i(1+\tau) & \cdots & x_i(1+(E-1)\tau) \\ \vdots & \vdots & \cdots & \vdots \\ x_i(h) & x_i(h+\tau) & \cdots & x_i(L) \\ \vdots & \vdots & \cdots & \vdots \\ x_i(L) & x_i(L+\tau) & \cdots & x_i(L+(E-1)\tau) \end{pmatrix} \begin{bmatrix} 0 \\ \vdots \\ 0 \\ \vdots \\ 1 \end{bmatrix} = \begin{pmatrix} x_i(1+(E-1)\tau) \\ \vdots \\ x_i(L) \\ \vdots \\ x_i(L+(E-1)\tau) \end{pmatrix}. \quad (13)$$

Based on Eqs. (11) and (12), we deduce a form (14)

$$\widehat{\psi}_i \begin{pmatrix} y_1(1) & y_2(1) & \cdots & y_E(1) \\ \vdots & \vdots & \cdots & \vdots \\ y_1(h) & y_2(h) & \cdots & y_E(h) \\ \vdots & \vdots & \cdots & \vdots \\ y_1(L) & y_2(L) & \cdots & y_E(L) \end{pmatrix} = \begin{pmatrix} x_i(1+(E-1)\tau) \\ \vdots \\ x_i(L) \\ \vdots \\ x_i(L+(E-1)\tau) \end{pmatrix}, \quad (14)$$

where $\widehat{\psi}_i(x) = \widehat{\phi}_i \psi_i(x)$. Eq. (14) suggests a mapping from the feature manifold to the final coordinate of the reconstructed manifold.

We add these contents to *Materials and methods* in the main text (see lines 498-511). To clarify the FRMM framework, we reorganize the processes in Fig.2 (d) and Eq. (4). The related code is free on <https://github.com/wt1234wt/FRMM-framework>.

Fig. 2 Sketch of FRMM framework. To forecast all components in an N -dimensional system (a), we find its E -dimensional ($E \ll N$) representations from delay embedding (b) and feature embedding (c). Thus, an N -dimensional dynamical system M is represented by two isomorphic low-dimensional manifolds (i.e., feature manifold M_0 and reconstructed manifold M_{x_i}). The foundation of an isomorphism suggests a one-to-one mapping between feature manifold M_0 and reconstructed manifold M_{x_i} . Then, it is possible to find a mapping $\tilde{\psi}_i$ ($i = 1, 2, \dots, N$) from the feature manifold M_0 to the final coordinate of the reconstructed manifold M_{x_i} . Therefore, the feature manifold M_0 can be utilized as a generalized predictor to find the future dynamics (purple elements) of all components in complex systems (d).

6. The author updated the colors of the points with blue curves. However, different colors of points in Fig. 6 are still being used without any explanation. I think the authors should pay more attention to these details and double-check them.

Response: Sorry for our mistakes. There are also no specific implications for different colors in Fig. 6. We replace it with a more natural one. We have checked the whole work again and modified all the shortages as much as possible.

Fig. 6 Performance with several ρ feature embedding techniques, including LLE (a), Laplacian (b), ISOMAP (c), and LTSA (d). LLE and Laplacian algorithms fail to find faithful low-dimensional representations of Lorenz attractor, resulting in poor performance for some components (e.g., variable z). While LTSA and ISOMAP preserve the fundamental geometry of the original attractor, FRMM yields reliable predictions for all components. Several algorithms can be utilized for feature embedding in the 3-dimensional Lorenz system, and an integration of Diffusion map performs the best predictions among them (Fig. 3(a)).

Reviewer #1 (Remarks on code availability):

The authors didn't provide the codes of the proposed method.

Response: Thanks for your suggestions. We have uploaded the main code and real-world datasets on <https://github.com/wt1234wt/FRMM-framework>

Reviewer #2 (Remarks to the Author):

The authors have revised the paper nicely, I am happy to recommend it for publication in Nat Comm.

Response: Many thanks for your recommendation as well as your helps to our article.

REVIEWERS' COMMENTS

Reviewer #1 (Remarks to the Author):

A minor suggestion:

Some most recent relevant works that apply the STI-based prediction method to real-world scenarios should be cited.

---Earthquake alerting based on spatial geodetic data by spatiotemporal information transformation learning. Proceedings of the National Academy of Sciences, USA, 2023, 120(37):e2302275120.

Reviewer #1 (Remarks to the Author):

A minor suggestion:

Some most recent relevant works that apply the STI-based prediction method to real-world scenarios should be cited.

---Earthquake alerting based on spatial geodetic data by spatiotemporal information transformation learning. Proceedings of the National Academy of Sciences, USA, 2023, 120(37):e2302275120.

Response: Thanks for your suggestion, we have cited this article in our work, see Ref. 44.